



# Modeling seasonal-to-decadal ocean-cryosphere interactions along the Sabrina Coast, East Antarctica

Kazuya Kusahara[1], Daisuke Hirano[2, 3], Masakazu Fujii[2, 3], Alexander D. Fraser[4], Takeshi Tamura[2, 3], Kohei Mizobata[5], Guy D. Williams[6], Shigeru Aoki[7]

5  [1]Japan Agency for Marine-Earth Science and Technology (JAMSTEC), Yokohama, Kanagawa, 236-0001, Japan
[2]National Institute of Polar Research, Tachikawa, Tokyo, 190-8518, Japan
[3]Graduate University for Advanced Studies (SOKENDAI), Tachikawa, Tokyo, 190-8518, Japan
[4]Australian Antarctic Program Partnership, Institute for Marine and Antarctic Studies, University of Tasmania, nipaluna/Hobart, Tasmania, 7004, Australia
10  [5]Department of Ocean Sciences, Tokyo University of Marine Science and Technology, Tokyo, 108-8477, Japan
[6]Marine Solutions Tasmania, Newtown, Tasmania, 7008, Australia
[7]Institute of Low Temperature Science, Hokkaido University, Sapporo, Hokkaido, 060-0819, Japan

*Correspondence to*: Kazuya Kusahara (kazuya.kusahara@gmail.com, kazuya.kusahara@jamstec.go.jp)

List of ORCID

Kazuya Kusahara          : 0000-0003-4067-7959

Daisuke Hirano           : 0000-0002-8047-1544

Masakazu Fujii           : 0000-0003-0527-1742

20   Alexander D. Fraser     : 0000-0003-1924-0015

Takeshi Tamura           : 0000-0001-8383-8295

Kohei Mizobata           : 0000-0001-7531-2349

Guy D. Williams          : 0000-0002-3975-2977

Shigeru Aoki             : 0000-0002-3314-484X



**Abstract**

The Totten Ice Shelf (TIS) and Moscow University Ice Shelf (MUIS), along the Sabrina Coast of Wilkes Land, are the floating seaward terminuses of the second-largest freshwater reservoir in the East Antarctic Ice Sheet. Being a marine ice sheet, it is vulnerable to the surrounding ocean conditions. Recent comprehensive oceanographic observations, including bathymetric measurements off the Sabrina Coast, have shed light on the widespread intrusion of warm modified Circumpolar Deep Water (mCDW) onto the continental shelf and the intense ice-ocean interaction beneath the TIS. However, the spatiotemporal coverage of the observation is very limited. Here, we use an ocean–sea ice–ice shelf model with updated bathymetry to better understand the regional ocean circulations and ocean-cryosphere interactions. The model successfully captured the widespread intrusions of mCDW, local sea-ice production and the ocean heat and volume transports into the TIS cavity, facilitating an examination of the overturning ocean circulation within the cavities and the resultant ice-shelf basal melting. We found notable differences in the temporal variability of ice-shelf basal melting across the two adjacent ice shelves of the TIS and the western part of the MUIS. Ocean heat transport by mCDW controls the low-frequency interannual-to-decadal variability in ice-ocean interactions, but the sea-ice production in the Dalton Polynya strongly modifies the signals, explaining the regional difference between the two ice shelves. The formation of a summertime eastward-flowing undercurrent beneath the westward-flowing Antarctic Slope Current is found to play an important role in the seasonal delivery of ocean heat to the continental shelf.



## 1 Introduction

Satellite-based observations have revealed that the Antarctic Ice Sheet has lost mass in recent decades (Rignot et al., 2019, 2008; Pritchard et al., 2012; Shepherd et al., 2018, 2012; Otosaka et al., 2022). This mass loss directly results in global sea-level rise, having a far-reaching effect not only on the physical and biological environment of Earth but also on human society (Mimura, 2013). The Intergovernmental Panel on Climate Change (IPCC) pointed out in the Sixth Assessment Report that the negative mass balance of the Antarctic Ice Sheet will become the major contributor to future sea-level rise in the coming
centuries, albeit with large uncertainties in the magnitude (IPCC, 2021). The Antarctic Ice Sheet gains mass by snow accumulation over the Antarctic continent and loses it mostly through two ablation processes at the surrounding ice shelves: basal melting at ice-shelf bases and iceberg caving at ice-shelf fronts (Greene et al., 2022). The total amount of the Antarctic ice-shelf basal melting has been estimated to be greater than that of iceberg calving (Rignot et al., 2013; Depoorter et al., 2013; Liu et al., 2015). Because the Antarctic ice-shelf base is the place where the Southern Ocean directly interacts with the Antarctic
Ice Sheet, oceanographic conditions along the Antarctic coastal margins are very important for understanding ice-ocean interactions linked to global sea-level rise.

Regions of the Antarctic Ice Sheet where the bedrock is below sea level are called marine ice sheets. The bedrock of marine ice sheets in coastal regions tends to become deeper further inland. It is known that ice sheet/ice shelf systems on such
retrograde slopes are more vulnerable to warm water intrusion into ice-shelf cavities. This type of topography, which serves as a fundamental precondition for a phenomenon known as Marine Ice Sheet Instability (Weertman, 1974), causing grounding line retreat and eventually destabilizing the inner ice sheets. Note that multiple factors, including the retrograde slope, basal boundary conditions, ice sheet geometry, bed curvature, and ice-climate feedbacks, collectively contribute to the potential occurrence of the instability (Sergienko, 2022). In the present-day Antarctic Ice Sheet, marine ice sheets are found extensively
in the West Antarctic Ice Sheet and in eastern parts of the East Antarctic Ice Sheet (the Aurora Subglacial Basin and the Wilkes Subglacial Basin), where rapid ice-sheet mass loss by marine ice sheet instability could potentially occur (Morlighem et al., 2020; Fretwell et al., 2013). Many studies have focused on the West Antarctic Ice Sheet, and there is a growing consensus that intrusions of warm Circumpolar Deep Water (CDW) onto continental shelf regions have eroded the West Antarctic ice shelves and grounding lines in recent decades (Payne et al., 2004; Shepherd et al., 2004; Kimura et al., 2017; Milillo et al., 2019;
Naughten et al., 2022). Similar potentially unstable conditions have also been recently noted in the East Antarctic Ice Sheet due to improved bedrock topography (Fretwell et al., 2013; Morlighem et al., 2020) and advances in ice-sheet modeling. One such area is the Totten Glacier (TG) in the Aurora Subglacial Basin, which has the seaward termination of the Totten Ice Shelf (TIS). Several ice-sheet modeling studies of future and past warm periods have indicated that the mass loss from the ice sheet in the Aurora Subglacial Basin substantially contributes to global sea-level rise, as well as the mass loss from the West
Antarctic Ice Sheet (Golledge et al., 2015; Pollard et al., 2015; DeConto and Pollard, 2016).

The TG is the second largest freshwater reservoir in the East Antarctic Ice Sheet, storing ice that is equivalent to more than 3.5 m of global sea-level (Greenbaum et al., 2015; Li et al., 2015; Young et al., 2011; Morlighem et al., 2020). A substantial amount of ice is discharged from the glacier/ice sheet across the grounding line, estimated to be from 69.0–72.6 Gt yr$^{-1}$ for the
period 1979–2017 (Rignot et al., 2019). The long-term mass balance of the TG over the period, estimated from a combination of satellite records and a regional atmospheric climate model, has shown a cumulative mass loss of 236 Gt (Rignot et al., 2019). Several satellite observational studies over different analytical periods have reported a similar acceleration in the regional ice discharge (Li et al., 2016; Rignot et al., 2019). However, a recent study by Miles et al. (2022) showed a decreasing trend after the discharge peaked around 2010. The discrepancies between the conclusions of these studies are likely due to different
periods and lengths of the analyses, indicating the presence of strong interannual variability and the need for long-term analysis to more robustly capture the changes. Variations in the ice velocity of the TG are tightly synchronized with changes in the



glacier thickness and the grounding line position. An increase in the ice velocity/discharge is accompanied by thinning in the coastal area and grounding line retreat, and vice versa (Li et al., 2015, 2016; Miles et al., 2022). Previous observational and modeling studies have inferred that the interannual variability in glacier/ice sheet variable is strongly controlled by changes in

surrounding ocean temperatures (Roberts et al., 2018; McCormack et al., 2021), in particular through intrusion of warm modified CDW (mCDW) onto the continental shelf region. A modeling study using an asynchronously coupled ice-ocean model to project future changes in the TG and TIS also suggested that an increase in mCDW in a warming climate will accelerate the mass loss and widespread retreat of the local grounding line (Sun et al., 2016; Pelle et al., 2021). Note that a recent study by Pelle et al. (2020) has indicated, however, that the Totten Glacier may be less susceptible to Marine Ice Sheet

Instability than previously thought, as the bedrock slope is prograde for approximately 10 km upstream of the current grounding line. However, this topographic configuration doesn't diminish the importance of changes in oceanic conditions on the glacier's sensitivity.

The coastal margin off the Sabrina Coast had few ocean observations until recently. Before an Australian oceanographic survey

conducted in January 2015 (Rintoul et al., 2016; Silvano et al., 2017), there were only two CTD stations just south of the shelf break (Bindoff et al., 2000; Williams et al., 2011). Although these isolated CTD profiles showed the local presence of warm bottom-intensified mCDW on the continental shelf break, the details of their potential pathways across the shelf and roles in ocean-cryosphere interaction were unclear at the time. Presenting ground-breaking new CTD and bathymetry data in front of the TIS, Rintoul et al. (2016) confirmed for the first time the direct access of warm mCDW toward the TIS through deep

troughs. Silvano et al. (2017) used the extended oceanographic observations to show that mCDW is abundant in deep layers over the continental shelf region off the Sabrina Coast. The mCDW is transported from offshore across the shelf break (Hirano et al., 2021), and the locations of the intrusions are consistent with the southward flow of quasi-stationary cyclonic eddies formed over the continental slope and rise (Mizobata et al., 2020). While bathymetry plays a vital role in regulating oceanic transport across the shelf break, over the continental shelf, and into ice-shelf cavities, there is unfortunately a lack of precise

regional bathymetry data in this area. Although recent oceanographic and geophysical studies have updated the available bathymetric data, the improvements have been somewhat piecemeal: Nitsche et al. (2017) detected wide and deep troughs within the shelf break, Silvano et al. (2019) pointed out deeper depression in the continental shelf than the previous datasets had shown, and Greenbaum et al. (2015) derived ocean bathymetry under the TIS.

Several numerical modeling studies have been conducted to better understand basal melting at the TIS and the surrounding ocean (Gwyther et al., 2014, 2018; Khazendar et al., 2013; Nakayama et al., 2021; Van Achter et al., 2022). However, as mentioned above, there remains large uncertainties in the regional bathymetry. In addition, the selection of bathymetry in past modeling studies was largely left to the discretion of the modeler, e.g., the use of the most recent available at the time, the modelers' preference, or constrained by their model configuration. Due to the bathymetric uncertainties, the previous studies

could only focus on individual processes in the overall ocean-cryosphere system, such as mCDW intrusion, coastal water mass transformation in polynyas, or ice-shelf basal melting at the TIS. While the previous modeling studies all determined the presence of warm mCDW inflow onto the continental shelf region, but there were varying representations of its spatial patterns. It has been suggested that the inflow of mCDW onto continental shelf regions is related to the Antarctic Slope Front/Current (ASF/ASC) system on the upper continental slope region (Nakayama et al., 2021; Thompson et al., 2018; Silvano et al., 2019)

and that sea-ice production in local coastal polynya plays a vital role in distinct water mass transformations over the continental shelf (Gwyther et al., 2014; Khazendar et al., 2013). It should be kept in mind that the results in these previous regional models were heavily influenced by their artificial lateral boundaries, the conditions which were often derived from a different coarse-resolution ocean model. Furthermore, the model integration periods (10–20 years) used in many of the previous studies are too short to be used in a detailed examination of interannual variability.




In this context, a recent study by Hirano et al. (2023) has comprehensively described the regional ocean circulation of mCDW from the shelf break to the TIS, using new bathymetry, extensive oceanographic observations over the Totten continental shelf, and results from a high-resolution ocean model. Our study uses the same model to conduct a detailed examination of the seasonal, interannual, to decadal variability, complementing the modeling part in Hirano et al. (2023), which used the annual-

mean climatology. Together with the inclusion of the updated bathymetric data off the Sabrina Coast, the strengths of our numerical model include the novel model configuration whose entire model domain is the Southern Ocean with a horizontal resolution within the focal region of less than 4 km (Fig. 1), and a long integration period over 70 years. These improvements allow us to represent regional ice-ocean interactions which were consistent with the recent observation (Hirano et al., 2023), and we can extend the discussion to longer interannual-to-decadal timescales. In Section 2, we describe the model

configuration and compare the bathymetric data used in the model with several existing bathymetric datasets. We then show model results for the sea-ice field (Section 3), the ice-shelf basal melting (Section 4), the ocean fields over the continental shelf, the ocean circulation within the ice shelf cavities (Section 5), and the ocean-cryosphere linkage between them (Section 6).

## 2 Model configuration

### 2.1 An ocean-sea ice-ice shelf model

This study utilized an ocean–sea ice–ice shelf model (Kusahara and Hasumi, 2013, 2014) previously employed in both regional and circumpolar Southern Ocean modeling studies (e.g., Kusahara 2017, Kusahara 2021). The model equations are solved in an orthogonal, curvilinear, horizontal coordinate system (Hasumi, 2006). The horizontal coordinate has two singular points,

and we placed the model's singular points on the same longitude (75°S, 120°E and 0°, 120°E) to regionally increase the horizontal resolution off the Sabrina Coast (Fig. 1a) while maintaining a circumpolar Southern Ocean of the model's domain. With this configuration, the focal region can be covered with a horizontal resolution of less than 4 km, allowing us to reasonably resolve the bathymetry and coastal boundaries of land, ice shelves, and fast ice (i.e., "icescape") in the model. An artificial northern solid boundary was placed at a latitude range of 43°S to 30°S, varying with longitude. Although the horizontal

resolution becomes coarser outside the region of interest, the model's grid configuration has a strong advantage in escaping the nontrivial issue of the east-west lateral boundary conditions of regional models in the Southern Ocean. Such methods have often been used in Antarctic coastal ocean modeling studies (Marsland et al., 2004; Kusahara et al., 2021). The ocean component uses the z-coordinate system with the following vertical grid spacing from the surface to the bottom: 4 levels of 5 m, 59 levels of 20 m, 30 levels of 40 m, and 31 levels of 100 m. Partial step representation is adopted for the bathymetry and

ice-shelf draft (Adcroft et al., 1997).

The sea-ice and ice-shelf components in the model are the same as those used in our previous study of regional ocean-cryosphere interactions in Lützow-Holm Bay (Kusahara et al., 2021). There are landfast-ice-covered regions along the Sabrina Coast, which characterizes the regional icescape (Fraser et al., 2012, 2021; Van Achter et al., 2022; Nihashi and Ohshima,

2015). It is important to take account of the landfast-ice distribution for reproducing sea-ice production in the model. In this study, observation-based areas of persistent landfast-ice cover (Fraser et al., 2012) were specified as thin ice-shelf grid cells with a constant thickness (5 m). As a first approximation, the spatial distribution of landfast ice was assumed to be constant throughout the model integration. We used observation-based coefficients of the thermal and salinity exchange velocities for the ice shelf ($\gamma_t$=1.0×10$^{-4}$ ms$^{-1}$, $\gamma_s$=5.05×10$^{-7}$ ms$^{-1}$, Hellmer and Olbers, 1989), and applied one-tenth coefficients for landfast

ice to consider the difference in the tidal speed between the ice-shelf cavity and the open ocean in the parameterization. In our





previous ocean modeling for Lützow-Holm Bay, we confirmed that the magnitude of landfast-ice melting did not significantly affect the variability of the inflow of mCDW onto the continental shelf (Kusahara et al., 2021).

### 2.2 Bathymetry off the Sabrina Coast

Previous ocean modeling studies of Antarctic coastal regions (Kusahara et al., 2021; Sun et al., 2022) have pointed out a general concern about the accuracy of bathymetry along the Antarctic coastal region, where bathymetric data are not well constrained due to the insufficient number of direct water depth observations. This is especially true for the region off the Sabrina Coast. Severe sea-ice conditions and a large area of permanent/multiyear landfast ice restricts access to these areas, which limits direct observations. As shown below, there are considerable differences between the regional bathymetry in

available datasets (Fig. 2), which reflects the large uncertainty. A recent observational study by Hirano et al. (2023) was successful in measuring key bathymetric features using a multibeam echo sounder and air-bone-based expendable sensors (Fig. 2a), and they compiled a regional bathymetry, blending their observations with GEBCO2020 (Fig. 2b). Figure 2c–f shows horizontal maps of the bathymetry off the Sabrina Coast in four different datasets: GEBCO2020 (GEBCO_2020 Grid), BedMachine Antarctica v2 (Morlighem et al., 2020), RTopo-2 (Schaffer et al., 2016), and ETOPO1 (Amante and Eakins,

185    2009).

Several numerical ocean modeling studies have been conducted on the Sabrina Coast, but they used different bathymetry datasets. In low-resolution ocean models (e.g., one degree) or regions where the water depth is well constrained, the differences between datasets would not be much of an issue in creating a model bathymetry. In high-resolution ocean modeling with a

resolution of several kilometers, it is vital to prepare reliable bathymetry in the focal region. Model bathymetry accuracy is directly linked to the reproduction of ocean circulation in the model. Particularly in the Antarctic coastal margins, poleward CDW intrusions from shelf breaks to continental shelves, which are strongly constrained by local bathymetry (Nitsche et al., 2017), play an important role in ocean-cryosphere interactions. In this study, we created a model bathymetry to combine the recently-compiled bathymetry over the Sabrina Depression (Hirano et al., 2023) and BedMachine near/underneath the ice

shelves. It should be noted that the ocean bathymetry in Bedmachine is a blend of IBSCO (Arndt et al., 2013) for the background and additional sources for the regional refinements (Morlighem et al., 2020). For the Totten ice shelf, the inversion product from Greenbaum et al. (2015) was used.

Here, we compare several bathymetric datasets with direct measurements and justify the model's bathymetry, with a focus on

the shelf break region and three regions over the continental shelf. The three selected regions are the eastern and central parts of the Sabrina Depression and a coastal region in front of the TIS, labeled with x, y, and z in Fig. 2b, respectively. Firstly, the shelf break in the observation and all datasets (e.g, 1000-m depth contour) are identified within a latitude range between 65.5°S and 65°S. The depth contours representing the shelf break in the observation, GEBCO2020, and ETOPO1 are found to smoothly extend in the east-west direction, whereas those in BedMachine and RTopo2 have unrealistic/artificial bumps along

the shelf break, which probably originated from gravity inversion processes. Secondly, the observed water depth in area x is shallower than 400 m, and it is thus an effective topographic barrier between the Sabrina Depression and east of 121°E. This bathymetric feature is captured in GEBCO2020 and RTopo-2, whereas in BedMachine and ETOPO1, there is a wide trough-like structure (> 500 m), which differs from the observed one. Thirdly, the central part of the depression is deeper than 800 m (see region y). GEBCO2020 and BedMachine represent the deep depression, although the overall structure of the depression

differs between the two. In RTopo2 and ETOPO1, the depression is shallower than the observed depth. Finally, looking at the region in front of the TIS (region z), the observed deep trough (> 600 m) that connects the SB and the TIS cavity (Hirano et al., 2023) is confirmed only in Bedmachine. The other three datasets show shallower structures toward the coast or ice front.



The recently-compiled bathymetry (Fig. 2b) reasonably captures the observed bathymetry in the Sabrina Depression. However, because the data does not include bathymetry below the ice shelf, we used BedMachine for the bathymetry in the ice shelf

regions. In the region within 20 km from the ice front, which is treated as a transition zone, we selected a deeper depth between the two. We consider that our model topography is the best estimate when taking into account the direct bathymetric observations.

## 2.3 Atmospheric conditions

The ocean-sea ice-ice shelf model was forced with atmospheric surface boundary conditions that include wind stresses, wind speed, air temperature, specific humidity, downwelling shortwave radiation, downwelling longwave radiation, sea-level air pressure, and freshwater flux. In this study, the daily surface boundary conditions were estimated from ERA5 datasets (Hersbach et al., 2020; Bell et al., 2021), using the bulk formula of Kara et al. (2000) for wind stresses and sensible/latent heat flux.


To understand atmospheric forcing, we calculated the seasonal climatology of sea-level pressure and surface wind averaged from 1981 to 2010 (Fig. 3). There are three low-pressure systems over the Southern Ocean, and the Sabrina Coast is located on the south-eastern side of the low-pressure system in the Indian Sector (see the red box in Fig. 3). Westerly and easterly winds prevail throughout the year to the north (equatorward of 63°S) and south (poleward of 64°S) of this low pressure system,

respectively. Note that the wind regime (negative wind stress curl) creates the Antarctic divergence zone where deep CDW can dynamically move upward to the subsurface due to surface Ekman divergence (Marshall and Speer, 2012; Tamsitt et al., 2017). In addition, the prevailing easterly wind in the coastal region is the driver of westward-flowing ocean currents along the Antarctic coastal regions, known as the ASF/ASC system (Thompson et al., 2018). The wind off the Sabrina Coast is directed west-northwest throughout the year, and reaches a maximum (> 5.5 m/s) in the autumn and winter seasons (from June

to August), which is consistent with the seasonal cycle of local atmospheric pressure. The wind speed in the spring and summer seasons is relatively mild, but the magnitude is larger than 3.0 m/s in the climatology.

## 2.4 Boundary/initial condition and spin-up integration

The model's initial oceanic properties of temperature and salinity were derived from January fields in the World Ocean Atlas

2018 (Locarnini et al., 2018; Zweng et al., 2019), and the velocity fields were filled with zero over the model domain. For the water properties in the ice-shelf cavities, ocean properties at the nearest oceanic grid were used. North of 40° S or near the artificial northern wall, the water properties were restored to the monthly climatology with a damping timescale of 10 day. In the region where the horizontal resolution is coarser than 4.2 km, the sea surface salinity (SSS) was restored to the monthly climatology to avoid unrealistically deep convection. There is no SSS restoring in the focal region. We ran the ocean–sea ice–

ice shelf model for 30 years with 1951–1960 forcing (e.g., three cycles), and subsequently performed a 3-year adjustment with the 1951-year forcing to obtain a spun-up initial condition for use in the historical experiment for the period 1951–2021("CTRL" case).

To monitor the spin-up of the ocean–cryosphere system off the Sabrina Coast, we use the time series of the total ice-shelf basal

melt amount in the region, as the sum of the TIS, eTIS, wMUIS, and MUIS (Fig. 4). This is a useful metric because the basal melting is controlled by the integrated effects from the interactions across local sea-ice processes, the large/regional ocean circulation, and coastal water mass formation. At the beginning of the first cycle, ice-shelf basal melting is very active, with a melting amount larger than 200 Gt yr$^{-1}$, but it decreases to 50 Gt yr$^{-1}$ within approximately five years. In the second half of



the first cycle, ice-shelf basal melting varies between 50 Gt yr$^{-1}$ and 100 Gt yr$^{-1}$. Active melting at the beginning is due to the

extrapolation of relatively warm water from the nearest ocean grids to ice-shelf cavities. A comparison of ice-shelf basal melting between the three cycles demonstrates that the model reaches quasi-steady conditions in ice-shelf basal melting and the relevant coastal processes after the long 30-year spin-up integration. As shown later, 1951 is a relatively cold year, and sea-ice production is accordingly high. In the three-year adjustment with the 1951-year forcing, the ocean conditions are adjusted into cold conditions (purple line in Fig. 4). Comparing ice-shelf melting in the spin-up experiments (second and third

cycles and 3-year adjustment) with CTRL case, the differences are evident in the first two years, but the subsequent interannual variability is similar among them.

We performed an additional experiment in which the model continued to be forced with the 1951 surface forcing to check for inherent model drift ("CKDRF" case). The integration period is 1951–2021. In this paper, we don't mention the results from

CKDRF case, but the results are shown in the figures to confirm that the results from CTRL case are not model drift.

### 3. Sea-ice extent and production

Sea ice strongly characterizes the ocean surface conditions over the Southern Ocean. The presence or absence of sea ice and the formation/melting processes play a vital role in regulating ocean surface exchanges of heat, salt, freshwater, and momentum

between the atmosphere and the ocean. Therefore, it is essential for this study to assess the reproducibility of sea-ice fields in the model before delving into the ice-ocean interaction. Here, we show the model's sea-ice representation, paying attention to the large-scale sea-ice extent and coastal sea-ice production off the Sabrina Coast. In particular, the coastal sea-ice production significantly impacts the temporal variations in coastal ocean conditions and ice-shelf basal melting, as shown in later sections.

Figure 5 shows the observed and modeled seasonal cycle of regional sea-ice concentration and extent from 108°E to 128°E. The monthly and seasonal averages were calculated from 1981 to 2010. Although sea-ice extent in the model tends to be underestimated throughout the year (Fig. 5i), the model reproduces seasonal changes in the spatial pattern of sea-ice concentration, capturing a large-scale east-west gradient and small-scale features controlled by the coast and ice-front line (Fig. 5a-h). Looking at the coastal regions, the model successfully represents the Dalton Polynya on the western side of the prolonged

landfast ice at approximately 122°E and the year-round high sea-ice concentration just north of the TIS. We calculated the regional sea-ice extent anomaly, defined as the monthly deviation from the monthly climatology for the reference period (1981–2010), to assess the interannual variation in the model (Fig. 6). In all months, there are significant correlations between the observations and the model. Particularly in the colder months from April to October, the correlations are higher than 0.8. This indicates that the model reasonably represents interannual variation as well as the seasonal cycle, providing us the

confidence to examine further seasonal-to-interannual interactions between the sea ice and the upper ocean.

Sea-ice production is an essential metric for the formation and transformation of Antarctic coastal water masses (Morales Maqueda et al., 2004). Sea ice forms during cold months when large amounts of heat are lost from the ocean due to intensive cooling by the atmosphere. Since sea ice can contain less salt than the ocean, excess salt is released into the ocean surface

during sea-ice formation (e.g., brine rejection). It should be noted that the sea-ice salinity in the model is assumed to be 5 psu. When sea ice forms, cold (surface freezing point), high salinity, and dense water is formed, causing an unstable ocean condition and subsequent local convection. The Dalton Polynya is formed over the eastern part of the Sabrina Depression (Tamura et al., 2008; Nihashi and Ohshima, 2015; Orsi and Webb, 2022). Figure 7 shows the horizontal distribution of annual sea-ice production in the model, representing the Dalton Polynya with high sea-ice production (up to 10 m yr$^{-1}$). The southern and





western parts of the Sabrina Depression show relatively low sea-ice production (< 3m yr⁻¹). The east-west contract of sea-ice production in this region is roughly consistent with the distribution described by satellite-based estimations (Tamura et al., 2008; Nihashi and Ohshima, 2015; Tamura et al., 2016; Nakata et al., 2021). The seasonal variation indicates active sea ice production from March to October, with a peak from May to July (Fig. 7b).

The mean sea-ice production over the Sabrina Depression (SD box in Fig. 7a) for the reference period(1981–2010) is 98.3 km³ yr⁻¹ with a standard deviation of 12.6 km³ yr⁻¹. The modeled sea-ice production fluctuates between 80 and 120 km³ yr⁻¹ from 1951–2021 (Fig. 7c). The sea-ice production in the Dalton Polynya estimated from surface heat flux calculation ranges from 35 to 51 km³ yr⁻¹ (Tamura et al., 2008; Nihashi and Ohshima, 2015; Tamura et al., 2016; Nakata et al., 2021), while that estimated from sea-ice divergence is around 197 km³ yr⁻¹ (Orsi and Webb, 2022). Although there is a wide range in the
observations, the sea-ice production reproduced in the model is within the range of these estimates (Table 1), demonstrating that the model reasonably represents the surface forcing through the coastal sea-ice formation. It should be noted that since the observational estimates from the heat flux calculation were only conducted in thin sea-ice areas, the total sea-ice production estimates tend to be smaller than those for the region over the Sabrina Depression.

**4 Ice-shelf basal melting**

The model represents intensive ice-ocean interaction at the base of the ice shelves along the Sabrina Coast (Fig. 8 and Table 2). In this study, we mainly use the climatology of ice-shelf basal melt amount and rate averaged over the reference period (1981–2010) to examine the magnitude and spatial distribution. While the main conclusion remains the same independent of the averaging periods, this period was chosen to coincide with the reference period for the sea-ice analysis and to examine the
long-term mean field to minimize the influence of interannual variations. For our convenience, the ice shelves were categorized into four groups from west to east based on their geographic location: TIS, eTIS (eastern TIS), wMUIS (western MUIS), and MUIS. High melting rates exceeding 10 m yr⁻¹ are found near the grounding line of all ice shelves (Fig. 8a). In particular, melt rates of 15 m yr⁻¹ or higher are reproduced over a wide area near the TIS grounding-line zone, indicating very intense ice-shelf basal melting. It should be noted that the satellite-based observation estimates for circumpolar mean Antarctic ice-shelf basal
melt rate are around 0.81±0.11 m yr⁻¹ (Depoorter et al., 2013) to 0.85±0.1 m yr⁻¹ (Rignot et al., 2013), which are much smaller than the regional melt rate modeled here.

Here we summarise the regional ice-shelf basal melt amount and rate in the model (Table 2) and compare it to the satellite-based estimates (Table 3, Adusumilli et al., 2020; Depoorter et al., 2013; Liu et al., 2015; Rignot et al., 2013). Table 2 shows
the values for two periods of the reference period (1981–2010) and the entire period of the model integration (1951–2021). In the model, the TIS has the highest ice-shelf basal melt rate at approximately 8 m yr⁻¹, followed by the eTIS melt rate with approximately 6 m yr⁻¹. The ice-shelf basal melt rate at the wMUIS is smaller than 5 m yr⁻¹, and that of the MUIS is larger than 5.6 m yr⁻¹. The modeled ice-shelf basal melting amount over the whole TIS (TIS+eTIS) and the whole MUIS (wMUIS+MUIS) are estimated to be approximately 48–51 Gt yr⁻¹ and 27–28 Gt yr⁻¹, respectively. The modeled ice-shelf
basal melting amount and rate for the whole TIS are slightly smaller than the observational estimates, which range from 59 Gt yr⁻¹ to 64 Gt yr⁻¹, while those for the whole MUIS are very comparable to the observational estimates, 26–28 Gt yr⁻¹. A recent observation of ice-shelf basal rate near the TIS grounding line by an autonomous phase-sensitive radio-echo sounder (ApRES) was estimated to be 22 m yr⁻¹, which was reported to be lower by about 40% than the satellite observations (Vaňková et al., 2021). The model shows ice-shelf basal melting at a rate higher than 20 m yr⁻¹ at the grounding line (Fig. 8a), consistent with





the ApRES estimate. Although both the model and satellite observations have their own biases in ice-shelf basal melting, we consider that the model adequately reproduces the active ice-shelf basal melting along the Sabrina Coast.

Next, we show the amplitude and months of the maximum and minimum to examine the seasonal cycle of the ice-shelf basal
melt rate (Fig. 8b–d). The amplitude was defined as the difference in basal melt rate between the maximum and minimum months. The seasonal amplitude is smaller than the annual-mean ice shelf basal melt rate, indicating active ice-ocean interaction throughout the year. The southern halves of the TIS and MUIS and the entire wMUIS have relatively strong seasonality with an amplitude larger than 4 m yr$^{-1}$, whereas the northern halves of the TIS and MUIS exhibit moderate seasonality. The wMUIS and MUIS ice shelves have the maximum in February-March and the minimum from July to
October. The seasonal amplitude of the ice-shelf melt rate over the TIS is widely heterogeneous from region to region, as also confirmed in seasonal variations of ice-shelf melt rate averaged every 25 km from the ice front (Fig. 9). In the region within 25 km of the ice front, there is a minor seasonal variation. In the 25–50 km region, the minimum is in April, and the maximum is in September-October. In the 50–75km region, the minimum is in April, and the maximum is in September–October. Near the groundling zone, the minimum is in May, and the maximum is in September–October. These results indicate the increased
seasonal amplitude and delayed seasonality with distance from the ice front.

The time series of the modeled basal melting amount at the four ice shelves are shown in Fig. 10 (blue and red lines for the monthly and annual averages, respectively). The time series in CTRL case substantially deviated from that in CKDRF case, confirming that the interannual variability in CTRL case is not due to artificial drift or inherent model variability. The temporal
variability at the TIS/eTIS and wMUIS/MUIS differs considerably, with low-frequency interannual-to-decadal variability predominating in the TIS regions and high-frequency seasonal variability dominating in the MUIS regions. Wavelet power spectra (Torrence and Compo, 1998) for modeled ice-shelf basal melting amount were calculated to quantify the differences in the temporal variability (Fig. 11). It should be noted that the power spectrum was normalized by variance to compare different time series with different magnitudes of variability, and was also divided by scales to rectify the wavelet power
spectrum (Liu et al., 2007). The normalized power spectrum is displayed in logarithmic form in Fig. 11, and thus positive values indicate where the variation is higher than one standard deviation. There is seasonal variation in the TIS ice-shelf basal melting, but it is not statistically significant for many years. Considerable variability is found in the 2–11 year period band, with intermittent statistically significant variability at around 8 years for the period 1960–2000. The general pattern of the power spectrum in the frequency-time domain of the ice-shelf basal melting at the wMUIS is similar to that of the TIS but
with a statistically significant one-year signal and non-significant interannual variability. These results indicate the ice shelves are subject to common low-frequency forcing and localized high-frequency forcing.

## 5 Ocean properties, current, and transports

The thickness of the ice shelves along the Sabrina Coast ranges from tens to a few hundred meters at the ice shelf's edge to
several hundred meters to over 1000 meters towards the grounding line. The ice shelf is a thermal insulator between the ocean and the atmosphere, and the lateral flow of warm (relative to depth-dependent freezing point) ocean water into the ice-shelf cavities is the main cause of ice-shelf basal melting. The inflow is linked with both the local water mass transformation and the regional ocean circulations over the continental shelf and slope regions. In this section, in order to understand the regional ocean-cryosphere interactions along the Sabrina Coast, we present vertical and spatial distributions of ocean properties and
velocity and calculate ocean volume/heat transport across some key sections. The model results are presented in the following





order: near the ice front and under ice shelves (Section 5.1), over the Sabrina Depression (Section 5.2), and around the shelf break and upper continental slope region (Section 5.3).

### 5.1 Ice-front region and under the ice shelf

To begin with, we show vertical profiles of ocean temperature, salinity, and velocity along a section 5 km offshore from the Antarctic coast and ice front (Fig. 12). This figure is the annual-mean fields averaged over the reference period. Signals of warm, high salinity, and relatively dense water are identified in deep troughs in front of the ice shelves. As shown later, this warm deep water near the coastline and ice-shelf front originates from intrusions of mCDW onto the continental shelf across the shelf break. Along this section, the mCDW signal is clearly seen in the density layers denser than 27.6 kg m$^{-3}$. The velocity

field shows clearly that mCDW is trapped in the deep troughs, and warm deep water flows toward the ice-shelf cavities, with velocity magnitudes above 5 cm s$^{-1}$. The warm deep water signals are separated by a shallow bank located zonally between 119.5° E to 122° E, indicating less water exchange in the denser layers. Compensating flows are found in the upper layers lighter than 27.6 kg m$^{-3}$, reflecting a part of the typical ocean circulation formed in ice-shelf cavities.

Next, we examine seasonal variations in volume and heat transports (referenced to −2.0° C) into each ice-shelf cavity across the ice-shelf fronts (Fig. 13). We used the monthly-mean outputs of the ocean water properties and velocities for this calculation. The volume transports and the mean temperatures were calculated for each 0.02 kg m$^{-3}$ bin of the potential density for the inflow and outflow components. The upper panels in Fig. 13 show the net inflow volume transports and the mean temperatures of the inflow component. The annual-mean total volume transports into the TIS, eTIS, wMUIS, and MUIS are estimated to be

190, 48, 104, and 108 mSv, respectively. There is a sizable inflow of warm waters with temperatures higher than –1.0°C in the densest layers (e.g., near the seafloor, see also Fig. 12), with the seasonal peaks occurring from May to September in the TIS and eTIS. The potential density boundary of 27.6 kg m$^{-3}$ is shown in the figure as the green line for ease of comparison with other figures. The wMUIS reaches its peak between April and May, while the MUIS reaches its peak between October and December. Ocean heat transports into the cavities show a similar seasonality (lower panels in Fig. 13). The annual-mean

heat transports into the TIS, eTIS, wMUIS, and MUIS are estimated to be 723, 126, 331, and 331GW, respectively. According to these analyses, the deep trough in front of the TIS cavity (Figs. 1b and 12) serves as an effective heat conduit from the Sabrina Depression to the TIS cavity. The wMUIS's volume and heat transports show characteristic seasonal changes. From June to October, there are signals of cold water inflow in the relatively less-dense water masses, reflecting the cold water formation in the nearby coastal polynya during cold months. The existence of the seasonal cold water controls local heat

transport into the cavity.  A similar signal of seasonally-formed cold water is observed at the TIS inflow during September and November, but its magnitude is small. Because the MUIS's front is covered with landfast ice, there is no such seasonal variation, and the volume and heat transports by deep warm water become large during October-December.  The volume and heat transport analyses demonstrate that the timing of deep warm water inflow differs from one ice-shelf cavity to another.

We can extend the volume transport analysis described above to a vertical stream function by defining a semi-closed domain, which allows us to depict the ocean thermohaline circulation beneath an ice shelf (Fig. 14). It should be noted that this is the first modeling study to present the overturning circulation within the ice-shelf cavities with realistic bathymetry and ice-shelf draft (Hirano et al., 2023). We defined analysis sections in a semi-closed domain every 5 km from the ice front and performed the volume transport analysis (integrating the net volume transport laterally and from the dense to light water masses). Because

solid lateral boundaries are required to form the semi-closed domain, the TIS and eTIS are combined for this analysis. The mean potential temperature averaged over the model grids between the sections is also shown in the figure. The three regions (TIS+eTIS, wMUIS, and MUIS) have clear overturning circulations with deep warm water inflow and less dense cold outflow.



The TIS+eTIS overturning circulation has about 200 mSv inflow across the ice shelf front, with a maximum value of over 240mSv at a distance of 25 km poleward from the ice front. The density of the inflow ranges from 27.54 to 27.78 kg m$^{-3}$ at the ice-shelf front, but the bottom density of the inflow gradually decreases toward the inside, reaching 27.62 kg m$^{-3}$ near the grounding line. The overturning circulation under the wMUIS is relatively narrow in length, and the opposite circulation forms in front of the ice shelf, partially reflecting cold, relatively dense water formation during cold months. The MUIS has a similar overturning circulation to the TIS-eTIS but with a much smaller magnitude.

### 5.2 Sabrina Depression

The analysis in the previous subsection indicates that warm deep water inflow into the ice-shelf cavities predominantly regulates the seasonal variation of the regional ice-shelf basal melting (Figs. 8, 9, and 13). In this subsection, we take a closer look at seasonal variations in water mass exchange over the continental shelf to elucidate the water masses' linkage between the continental shelf region and ice-shelf cavities. Here, we examine seasonal variations of water mass transport and the ocean temperatures across the western, northern, and eastern boundaries of the SD box placed on the continental shelf (the inset in Fig. 15). The control volume, SD box, is the same one used in the sea-ice production analysis (Fig. 7). The largest inflow transport into the SD box occurs across the eastern boundary, with an annual average transport of 590 mSv (Fig. 15c); it peaks from March to May, reaching approximately 800–1000 mSv. The outflow transport from the eastern boundary is minor throughout the year, with an annual average of about 40 mSv. The largest outflow transport from the SD box is across the western boundary, with an annual average of 642 mSv (Fig. 15d). The seasonal variation in the outflow transport is similar to the inflow across the eastern boundary, with the maximum from March to May, approximately ranging from 800 to 950 mSv. Across the western and eastern boundaries, the inflowing and outflowing water masses have relatively light density classes (< 27.6 kg m$^{-3}$), and the water temperatures are relatively cold (< −1.6°C). Looking at the inflowing water masses from the northern boundary, although the annual-mean inflow transport is moderate (282 mSv), warmer water inflows (> −0.8°C) are identified in the dense water masses (> 27.6 kg m$^{-3}$). In the denser classes, the warmest water inflow occurs from October to January, and the water temperature is above 0°C (Fig. 15b). The annual-mean outflow transport from the northern boundary is 207 mSv, with the peak during March–May and the cold temperatures. This result clearly shows that warm deep water masses reaching the ice-shelf originate from water across the northern boundary of the SD box (i.e., the shelf break). It should be noted that the water balance for the SD Box is roughly balanced by the inflow and outflow transports from the three lateral boundaries, with the remaining 0.1% being explained by the sum of ice-shelf basal melting from the southern boundary (i.e, ice-shelf fronts), sea-ice production, transport, and melting over the SD box.

To examine in detail the spatial pattern and seasonal progression of warm deep water inflow onto the continental shelf across the shelf break, we present the annual-mean temperature averaged over depths deeper than 400 m (Fig. 16a) and a map of the month when the ocean temperature in each grid cell reaches the maximum (Fig. 16b). Furthermore, in Fig 17, we use vertical profiles to illustrate the seasonal progression of the potential temperature and potential density anomaly at selected four locations (shelf break, Dalton Polynya, and two coastal regions in front of the ice shelves, labeled with A–D in Fig. 16). Warm deep water crosses the shelf break at 117–120°E and flows south-eastward along the north-eastern flank of the Sabrina Depression (e.g., 600-m depth contour), forming a clockwise circulation. The warm deep water intrusion fills the entire Sabrina Depression with water warmer than −0.6°C. Near the coastal regions, although the ocean temperatures are colder than those in the central part of the depression, they are still above −1.0°C, and the warm signals extend toward the ice shelves along the deep troughs (see also Fig. 12 for the vertical profiles). Along the shelf break and upper continental slope regions, the ocean temperatures peak during December–January, while over the continental shelf, the ocean temperatures are the maximum from January to June, showing that the seasonal progression of the ocean temperatures differs greatly from one region to another.




At the shelf break region (Fig. 17a), the potential density surface of 27.7 kg m$^{-3}$ is present at around 300 m throughout the year with only small vertical variation, and there are warmer waters in deep layers, with a potential temperature range from −0.6°C to above 0°C. The ocean temperature in the deep layer reaches a maximum during November–February and a minimum during May–June. This seasonal progression at the shelf break explains the seasonal variation in the ocean temperature of inflow

transport in the dense water classes to the SD box across the northern boundary (Fig. 15). In the Dalton Polynya (Fig. 17b), a cold surface mixed layer develops during the fall and winter seasons when sea-ice production becomes active (Fig. 7). For example, the −1.6°C isotherm gradually deepens after April, reaching nearly 300-m depth by August. Below the seasonally-developed mixed layer, there is a year-round signal of the warm water intrusion along the seafloor. The months of maximum and minimum ocean temperatures are December–February and August, respectively, and the seasonal progression is one to

two months later than at the shelf break.

In front of the wMUIS (Fig. 17d), the seasonal progression is further delayed, with the maximum in May and the minimum in September–October. Here, the bottom ocean temperature in the maximum month is approximately −0.7°C and it is colder than that in the Dalton Polynya (Fig. 17b), reflecting successive water mass modification by interaction with the above cold mixed

layer along the pathway of the clockwise ocean circulation. In front of the TIS (Fig. 17c), the ocean temperature reaches a maximum in June–August and a minimum in February–March. These results indicate that there is a profound time lag of about half a year between the shelf break and coastal regions, reflecting the slow clockwise ocean circulation in Sabrina Depression. The lag is roughly accounted for by the timescale of the ocean circulation with typical current speeds of a few centimeters per second over the continental shelf. In both the coastal regions, although the thickness of the surface mixed layers in cold seasons

is greater by about 100 m than that in the Dalton Polynya, there are warm water signals with the potential to become heat sources for intensive ice-shelf melting.

### 5.3 Shelf break and upper slope regions

The analysis of inflow and outflow transport across the SD box's boundaries in Section 5.2 (Fig. 15) reveals that the intrusion

of warm deep water onto the continental shelf predominantly comes from the northern boundary. In this subsection, in order to investigate water mass exchange between the open ocean and the continental shelf regions, we place another control box over the shelf break and the upper continental slope regions (hereafter, "Slope Box") and examine the inflow and outflow of water mass transport across the lateral boundaries (Fig. 18). The western, eastern, and northern boundaries of the Slope Box were set to 117° E, 121° E, and the 2500-m depth contour, respectively. The Slope Box's southern boundary shares a part of

the SD box's northern boundary. We only calculated the inflow and outflow transports from the surface to 800 m, to focus on water mass exchange across the shelf break. On an annual-mean basis, there is a large inflow from the eastern boundary into the Slope Box, with the total transport over 1000 mSv (Fig. 18c). Balancing the inflow transport, outflow transports exceeding 400 mSv are observed at both the western and northern boundaries (Fig. 18d–e). This inflow/outflow pattern becomes more prominent from March to September, reflecting that the stronger easterly wind drives the westward-flowing ASC (Fig. 3) and

the ASC flows through the box. Interestingly, the inflow/outflow pattern from November to January differs markedly. Warm water inflows in the dense water classes come from the western and northern boundaries into the box and outflows from the eastern boundary, indicating the existence of seasonal eastward-flowing currents in subsurface layers (e.g., undercurrent). Focusing on dense water masses (>27.6 kg m$^{-3}$) in these months, there are inflow transports ranging from 100 mSv to 200 mSv with the ocean temperature reaching +0.4°C. As shown below, in the summer months, an eastward-flowing undercurrent

forms beneath the ASF/ASC, controlling the warm deep water intrusions across the shelf break onto the continental shelf region.



Figures 19 and 20 show the velocity fields in December, June, and September in the surface (0–100m) and middle depth (400–600m) layers, respectively. The westward-flowing ASC develops in the surface layer in June and September (Fig. 19b–c), and the signals are identified in the mid-depth layer (Fig. 20b–c). In December, the flow pattern along the shelf break is substantially different. There is relatively modest westward surface flow, and a broadly structured eastward-flowing undercurrent forms in the region between 500-m and 2000-m depth contours (Fig. 20a). Figure 21 shows the vertical profiles of east-west ocean velocity and temperature in December and September along the 117°E line. The vertical profile of the ocean velocity clearly delineates the outline of the eastward-flowing undercurrent, which forms in the 200m to 1000m depth range and meridionally from 65.2°S to 65°S, just north of the surface core of the ASC. The undercurrent has a maximum velocity of approximately 5 cm/s, and the core is located near a depth of 500 m. The depth of the undercurrent core is shallower than that of the central trough of the Sabrina Depression (see blue line in Fig. 21). In September, when the ASF is most developed, the westward flow extends from the surface to the seafloor over the shelf break and upper continental slope regions. The transports of the surface current and undercurrent are shown in Fig. 22. We calculated the transports of the surface current by integrating the westward transport in density layers lighter than 27.7 kg m$^{-3}$ and that of the undercurrent by integrating the eastward transport in density layers denser than 27.6 kg m$^{-3}$. The surface current transport becomes small from November to January and reaches its maximum in September. The undercurrent transport reaches its minimum in September, starts to increase from October, and reaches the maximum in November–December. It should be noted that although the eastward transport increases from April to July, this is not due to the structure of the undercurrent in the west-east direction, but instead due to the small-scale local circulation of the flows in from the western boundary and out to the northern boundary (Fig. 20b).

## 6 Causes of interannual variability in the regional ocean-cryosphere interactions

As seen in Figures 10 and 11, there is considerable low-frequency interannual variability (timescales longer than 5–7 years) in the ice-shelf basal melting along the Sabrina Coast. In this section, we examine relationships among the ocean-cryosphere variables in terms of interannual variability. Figure 23 presents the correlation coefficients between the variables of the sea-ice production over the Sabrina Depression ("SP" in the figure), the ocean heat transports across key sections ("HT"), the ocean overturning circulation beneath the ice shelves ("OVT"), and the ice-shelf basal melting ("BM"). Annual averages were used in this analysis. The lower left half of the plot shows the correlation coefficients between a pair of variables after taking a 7-year running mean to focus on low-frequency interannual-to-decadal variations, and the upper right half shows that after removing the running mean from the original annual-mean time series to focus on high-frequency year-to-year variations. The heat transports are the southward component across the northern boundary of the SD box ("nSD") and the inflow components across the ice fronts (short abbreviations of WM, eT, and TI representing wMUIS, eTIS, and TIS, respectively, in Fig. 23).

There are robust positive correlations between ice-shelf basal melting at the TIS and the neighboring ice shelves, especially in the low-frequency variability (eTIS: r=0.96 and wMUIS: r=0.90). The correlation with the eTIS remains high for the high-frequency variability (r=0.86), but it decreases to r=0.71 for the more distant wMUIS, indicating that a different factor contributes to the high-frequency year-to-year variability of ice-shelf basal melting at the wMUIS. The most notable feature of this figure is that the southward ocean heat transport across the SD's northern boundary has positive correlations with ocean heat transports towards ice-shelf cavities, the overturning circulation within the cavities, and the ice-shelf basal melting. This indicates that the fluctuation of warm deep water intrusions into the Sabrina Depression is a common driving force for the interannual variability of these components. In particular, they far exceed the 95% significance level in the low-frequency variability. Sea-ice production is negatively correlated with most of these components. In particular, in the high-frequency




variability, there is a significant negative correlation between ice-shelf basal melting and sea-ice production, indicating that the cold water formation associated with sea-ice production erodes some of the warm deep water inflow into the ice-shelf cavities. In summary, the fluctuation in warm deep water intrusions across the shelf break determines the low-frequency variability in ice-ocean interactions along the Sabrina Coast, whereas the high-frequency year-to-year variability is determined by a combination of the local sea-ice production and the warm deep water intrusions.

Seasonal variations in the southward ocean heat transport across the SD's northern boundary reveal that the intrusion of warm deep water intrusion onto the continental shelf regions occurs throughout the year (Fig. 15b), with the warmest water entering in summer. The analysis of inflow and outflow transports across the Slope Box's boundaries, which fuel the southward heat transport, revealed that the inflow pattern of warm deep water varied significantly depending on the season, with the warm deep water inflow in summer transported by the eastward-flowing undercurrent and that in the other seasons carried by westward flow in the well-developed ASF/ASC system. Therefore, here we use the two periods of November-January and March-September to investigate the interannual variability of the ocean currents over the upper continental slope causing warm deep water intrusions and their relationship with coastal wind (Fig.3), which would be the driving force of the ocean currents. The months of February and October were assumed to be transition months and were excluded from this analysis.

The southward heat transport timeseries (Fig. 24a) explain the TIS basal melting (Fig. 10), as seen in the very-high correlations between the two (Fig. 23). We found that the ocean heat transport from the western boundary to the Slope Box is negatively correlated with wind speed (r=−0.41, over the 95% significant level) from November to January. This means that when the summertime wind speed is low, an eastward-flowing undercurrent develops. In contrast, from March to September, when wind speeds are strong, the ASF develops from the surface to deep layers, and the overall westward-flowing current delivers the ocean heat from the eastern boundary to the Slope Box. In summary, the origins of ocean heat transport to the Sabrina Depression vary greatly depending on the season; interestingly, the responses of the eastward-flowing undercurrent and westward-flowing ASC in ASF to wind speed are totally opposite. The magnitude of the correlation coefficient between the southward ocean heat transport across the SD's northern boundary and wind speed (r=−0.32) is smaller than those for the two periods of the year (r=−0.41 and r=+0.44), reflecting the opposite responses of the two ocean currents to the wind variability. The correlation coefficient of the southward heat transport with the eastward heat transport by the undercurrent, r=+0.23, is higher than that with westward heat transport by the ASC (r=−0.09), indicating the fluctuation in the summertime undercurrent mainly regulates the interannual variability in the heat transport onto the Sabrina Depression and the subsequent ice-shelf basal melting. Finally, we calculated correlation coefficients with the Southern Annular Mode (SAM) index, an indicator of changes in atmospheric circulation over the Southern Ocean. The SAM index in this study was calculated from the difference in the normalized zonal mean sea level pressure at 65° and 40°S (Gong and Wang, 1999). Correlation coefficients between wind speed and the SAM index are −0.70 in November-January and −0.47 in March-September, which exceed the 95% significant level (r=0.23). This means that a positive value in the SAM index leads to weaker coastal winds. The correlation coefficients of the undercurrent and ASC with the SAM index were +0.28 and −0.42, respectively. As with the response to wind speed, the undercurrent and ASC responses to SAM are totally opposed.

## 7 Summary and Discussion

The study is the first to simulate and investigate comprehensive ocean-cryosphere interactions off the Sabrina Coast over 70 years (1951–2021), including the ocean heat transport from the shelf break and upper slope regions to the continental shelf, the regional ocean circulations and properties over the Sabrina Depression, the water mass exchanges across the ice-shelf fronts, the overturning circulations within the cavities, and the overall ice-shelf basal melting. Previous modeling studies have





focused on specific aspects contributing to the ice-ocean interactions in this region, whilst valuable, there has been little discussion about the relative importance of each aspect from the overall perspective of the regional ocean-cryosphere interactions. The study is novel through its use of updated bathymetry to configure an ocean-sea ice-ice shelf model (Figs. 1 and 2) for detailed analyses of seasonal variations in the sea-ice fields, ocean fields, and ice-shelf basal melting. Importantly, the model has a long-term integration spanning more than 70 years making the output suitable for examining seasonal,

interannual to decadal variability.

The Antarctic coastal region is subject to easterly winds throughout the year, with the maximum in winter and the minimum in summer (Fig. 3). This easterly wind is the driving force for the ASF/ASC system (Thompson et al., 2018). This model shows that the ASF is well developed from the surface to the mid-levels with the stronger ASC in winter when the easterly wind is

stronger (Figs. 19bc, 20bc, 21b, and 22). Conversely, during the summer months from November to January, when the coastal easterly wind is weak, an eastward-flowing undercurrent is clearly present at depths of 400–800m just beneath the westward-flowing ASC (Figs. 19a, 20a, 21a, and 22). The interannual variations in both current systems are significantly correlated with easterly wind speed, but their signs are opposite (Fig. 24). The model reproduces the year-round intrusion of warm deep water onto the continental shelf (Fig. 15b). Since the depth of the shelf break to the north of the Sabrina Depression is 500–600 m

deep, the wintertime ASC variation at mid-depth and the summertime undercurrent are important for regulating the warm deep water intrusion. The warmest water of the intrusion occurs during the summer season (Fig. 15), and it originates from the eastward-flowing undercurrent on the upper continental slope (Fig. 18). The warmest signals near the shelf break are slowly carried toward the coastal regions and the TIS by the clockwise ocean circulation formed over the Sabrina Depression (Figs. 16 and 17). The high sea-ice production of the Dalton Polynya (Figs. 5 and 7) can substantially alter the water masses at the

surface and medium depths, while the warm signals from the deep water intrusion remain present along the seafloor (Figs. 12 and 17). The warm signals can be also detected in the seasonal variations of the coastal water masses flowing into the TIS cavity (Fig. 13). The cold water masses associated with local sea-ice formation are clearly visible in the wintertime inflow into the wMUIS cavity, located south of Dalton Polynya, whereas warm deep water dominates the inflow at the other ice-shelf cavities throughout the year. This modeling framework is the first to illustrate ocean overturning/thermohaline circulation

within the ice-shelf cavities in a realistic configuration, with warm inflows in deep layers and outflows of lighter, colder water at shallow depths (Figs. 12 and 14). The seasonal variation in ice-shelf melting varies with distance from the ice-shelf front, and the TIS has its maximum from September to November near the grounding line (Figs. 8 and 9). The interannual variability of the TIS basal melting has a predominant periodic variability of about 5–11 years (Figs. 10 and 11). The mCDW intrusion onto the continental shelf region is the root cause of this low-frequency variability (Fig. 22). The mMUIS exhibits a similar

low-frequency variability, although the seasonal fluctuation caused by convection associated with the sea-ice formation dominates the local variability (Fig. 11).

Despite these promising results, it is important to point out several limitations of this study and the differences from the previous studies. Using the updated bathymetry, we constrained the regional bathymetry of the Sabrina Depression, particularly

the coastal regions in front of the ice shelves, which are crucial for representing inflows of warm deep water to the cavities. However, as shown in Fig. 2, there are still large uncertainties in the bathymetry over the entire continental shelf. Since the representation of bathymetry in a high-resolution ocean model is directly linked to the reliability of the modeled ocean currents and properties, it is hoped that these regions of uncertain data are quickly resolved by future surveys. The present model has a horizontal resolution of 3–4 km for the target area (Fig. 1), and we were able to reproduce the bathymetry-guided ocean flow

patterns from the shelf break to the continental shelf region and ice-shelf regions. It should be noted that the raw bathymetry data have smaller structures within the deep troughs ranging from several hundred meters to 1 km, and such small-scale bathymetric features that cannot be resolved in this model may have a significant impact on the modeled ocean flow patterns.





There is also limitation regarding the bathymetry under the ice shelf in this model, and very recent research by Vaňková et al. (2023) has underscored the influence of under-ice-shelf bathymetry on ice-shelf basal melting. For instance, the accuracy of
the bathymetric data underneath ice shelves can greatly affect the representation of circulation under the ice shelf and the associated melting patterns. Incorporating the updated datasets, e.g., the inversion bathymetry from Vaňková et al. (2023) for grounding line zones, into numerical models at each stage could further refine our understanding of ocean-ice interactions, potentially revealing more nuanced basal melt patterns and circulations beneath the ice shelves. This is a promising direction for future research, which will contribute to a more detailed and accurate understanding of these complex ocean-cryosphere
interactions.

While the results of this study are consistent with previous works linking the magnitude of mCDW intrusions to the strength of the wind-driven ASF/ASC in the surface layer, but this study elucidates a different mechanism. Previous studies have proposed that the thickness and transport of the mCDW are determined in response to the vertical fluctuation of the lower
density surface of the ASF, e.g., the weaker the ASF, the more mCDW intrusion (Nakayama et al., 2021). The present model results shows that the density variability near the shelf break regions is limited to shallow depths ($< 200$ m) and that temperature changes in the deep layers are not linked to the vertical density fluctuation (see the potential density of 27.7 kg m$^{-3}$ in Fig.17a). Instead, we find that seasonal variations in the eastward and westward flows at mid depths over the upper continental shelf (Fig. 18) are responsible for the mCDW intrusion onto the Sabrina Depression (Fig. 15). This discrepancy among the models
may be related to whether or not the eastward-flowing undercurrent is reproduced. We consider that the present model result is feasible for the following reasons. From the observational perspective, eastward-flowing undercurrents beneath the ASF are often detected along the Antarctic coastal margins (Chavanne et al., 2010; Heywood et al., 1998; Núñez-Riboni and Fahrbach, 2009; Walker et al., 2013; Peña-Molino et al., 2016). According to Peña-Molino et al. (2016), mooring observations at 113°E revealed that the eastward-flowing undercurrent forms beneath the westward-flowing ASF in the East Antarctic coastal region.
From modeling perspective, several recent studies (Silvano et al., 2022, 2019; Kusahara et al., 2021; Assmann et al., 2013; Smedsrud et al., 2006) have demonstrated that representing the undercurrent on the upper continental slope plays a crucial role in mCDW intrusion and ice-shelf basal melting. Note that while the undercurrent disappears seasonally in the East Antarctic region, it persists year-round in the West Antarctic region (Silvano et al., 2022).

Given that the origin of the warm deep water flowing into the Sabrina Depression varies seasonally and that the possible warm water sources are widely distributed east to west over the continental slope and rise, it is therefore essential to understand how the ocean conditions are set up over the entire large area off the Sabrina Coast. The Totten area is located in the eastern part of the Kerguelen gyre in the Australian Antarctic Basin, where the Antarctic Circumpolar Current (ACC) deflect southward, bringing warmer offshore waters closer to the coastal regions. The interaction between the two strong currents (ACC and
ASF/ASC) and the rugged seafloor topography forms steady cyclonic eddies over the continental rise, which play a role in the north-south ocean heat transport between the deep and coastal oceans (Wakatsuchi et al., 1994; Mizobata et al., 2020; Hirano et al., 2021). The combination of the ACC location and the cyclonic eddy chain is thought to have a widespread impact on the ocean conditions over the continental slope and rise regions.

Recent studies from hydrographic and satellite observations imply a long-term southward migration of the ACC fronts in the Australian Antarctic Basin, which may promote more warm water toward the coastal regions (Yamazaki et al., 2021; Herraiz-Borreguero and Naveira Garabato, 2022). Linear trends of the model's TIS ice-shelf basal melting and the ocean heat transport to the Sabrina Basin over the entire integration period are positive and do not contradict these observations (Fig. 10). However, to fully investigate this proposed mechanism of the interaction between the offshore conditions and ice-shelf basal melting,
numerical experiments with longer integration periods or experiments in which the ACC front locations in the model are





artificially modified would be required. Furthermore, better knowledge of the role of the cyclonic eddies over the continental slope and rise regions, which directly connect deep ocean and continental slope regions, is also necessary. Here, we briefly conjecture the ocean responses by the major atmospheric mode, SAM. It is well known that the SAM index has a positive trend over the decades, and the trend is expected to continue in a global warming condition (Bracegirdle et al., 2013; Zheng et

al., 2013). The positive SAM indicates stronger eastward wind components over the Southern Ocean, leading to stronger westerly wind over the ACC region and weaker coastal easterly wind. The plausible changes in wind fields may cause more southward ocean heat transport by ACC and a stronger undercurrent in summer, causing more active ice-ocean interaction along the Sabrina Coast.

Although it has been recognized that the Antarctic slope undercurrent plays a key role in water mass exchanges between the deep and coastal oceans, its dynamics are not yet fully understood. Based on the results of this study, the bathymetry and seasonal variations in ocean stratification and alongshore wind are expected to be responsible factors for the formation. To investigate the dynamics of the slope undercurrent in more detail, numerical experiments in which these components are simplified or idealized would be helpful. Once the undercurrent dynamics are understood, it will be feasible to determine where

and how the undercurrent changes will occur in an on-going warming world. This will lead to a better understanding of not only changes in the inflow of warm deep water onto the continental shelf but also changes in the Antarctic ice-shelf basal melting, inner ice-sheet mass balance, and sea-level rise.



**Acknowledgments**:

This study was supported by JSPS KEKENHI Grants(JP19K12301, JP17H06323, JP20H04961, JP20H04979, JP21H04918, JP21H04931, JP21H03587, JP22H05003, and JP22H01337). KK was supported by MEXT-Program for the advanced studies of climate change projection (SENTAN) Grant Number JPMXD0722681344 and the Grant for Joint Research Program of the Institute of Low Temperature Science, Hokkaido University(23G021). DH and TT were supported by the Science Program of

Japanese Antarctic Research Expedition (JARE) as Prioritized Research Project (AJ0902: ROBOTICA and AJ1003: HeatCross), National Institute of Polar Research (NIPR) through Project Research KP-303, and the Joint Research Program of the Institute of Low Temperature Science, Hokkaido University, and TT was also supported by the Center for the Promotion of Integrated Sciences of SOKENDAI. MF was supported by the Science Program of Japanese Antarctic Research Expedition (JARE) and "Challenging Exploratory Research Projects for the Future" grant from ROIS (Research Organization of

Information and Systems). AF was supported by grant funding from the Australian Government as part of the Antarctic Science Collaboration Initiative program.

**Author contributions:** K.K led this study by performing all the numerical experiments and analyses and preparing the manuscript. All authors discussed the results and comments on the manuscript.


**Data availability:**
Compiled bathymetry: DOI: 10.17632/zf6h9pvxd8.1 (It will be available after acceptance.)
GEBCO2020: https://www.gebco.net/data_and_products/gridded_bathymetry_data/gebco_2020/
BedMachine: https://nsidc.org/data/nsidc-0756/versions/2

Rtopo2: https://doi.pangaea.de/10.1594/PANGAEA.856844
ETOPO1: https://www.ngdc.noaa.gov/mgg/global/global.html
WOA18: https://www.ncei.noaa.gov/access/world-ocean-atlas-2018/
ERA5: https://www.ecmwf.int/en/forecasts/datasets/reanalysis-datasets/era5
All numerical experiments were conducted using COCO (https://ccsr.aori.u-tokyo.ac.jp/~hasumi/COCO/) with an ice-shelf c

omponent (Kusahara and Hasumi, 2013), and numerical results for ice-shelf basal melting in this study will be available in a data repository (DOI: 10.17632/zf6h9pvxd8.1).

**Competing interest:** The authors declare no conflict of interest.




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



**Tables and Figures**


**Table 1:** Comparison of sea-ice production off the Sabrina Coast with previous studies.

|  | Sea-ice production (km$^3$/yr) | Period |
|---|---|---|
| This study | 98.3 ± 12.6 | 1981–2010 |
|  | 100.6 ± 13.1 | 1951–2021 |
| Tamura et al. (2008) | 42.6 ± 6.7 | 1992–2001 |
| Nihashi and Ohshima (2015) | 35 ± 4 | 2003–2010 |
| Tamura et al. (2016) | 51 ± 10 | 1992–2013 |
| Nakata et al. (2021) | 42 ± 4 | 2003–2010 |
| Orsi and Webb (2022) | 197.4 | 2003–2015 |




**Table 2:** Annual mean ice-shelf basal melt rate (m yr$^{-1}$) and amount (Gt yr$^{-1}$) for two periods (1981–2010 and 1951–2021). TIS indicates Totten Ice Shelf, MUIS Moscow University Ice Shelf, eTIS eastern TIS, wMUIS western MUIS.

| Ice shelf | Area (km$^2$) | 1981-2010 | | 1951-2021 | | |
|---|---|---|---|---|---|---|
| | | Amount (Gt/yr) | Rate (m/yr) | Amount (Gt/yr) | Rate (m/yr) | |
| TIS | 5581 | 41.7 ± 5.8 | 8.1 ± 1.1 | 38.9 ± 7.8 | 7.6 ± 1.5 | |
| eTIS | 1651 | 9.1 ± 1.3 | 6.0 ± 0.9 | 8.7 ± 1.6 | 5.8 ± 1.0 | |
| total TIS | 7232 | 50.8 ± 7.0 | 7.7 ± 1.1 | 47.7 ± 9.3 | 7.2 ± 1.4 | |
| wMUIS | 1295 | 5.7 ± 1.1 | 4.8 ± 0.9 | 5.5 ± 1.2 | 4.7 ± 1.0 | |
| MUIS | 4219 | 22.2 ± 2.7 | 5.7 ± 0.7 | 21.8 ± 9.3 | 5.6 ± 0.7 | |
| total MUIS | 5514 | 27.9 ± 3.6 | 5.5 ± 0.7 | 27.3 ± 3.7 | 5.4 ± 0.7 | |






**Table 3:** Observation-based annual mean basal melt rate (m yr$^{-1}$) and amount (Gt yr$^{-1}$) of the total TIS and the total MUIS.

| Total TIS | Amount (Gt/yr) | Rate (m/yr) | Period |
|---|---|---|---|
| Rignot et al. (2013) | 63.2 ± 4 | 10.5 ± 0.7 | 2007-2008 |
| Depoorter et al. (2013) | 64 ± 12 | 9.89 ± 1.92 | 2007-2009 |
| Liu et al. (2015) | 63 ± 5 | 14.1 ± 1.2 | 2005-2011 |
| Adusumilli et al. (2013) | 64.0 ± 11.0 | 11.5 ± 2.0 | 1994-2018 |
| | 59.4 ± 11.0 | n/a | 2010-2018 |
| Total MUIS | Amount (Gt/yr) | Rate (m/yr) | Period |
| Rignot et al. (2013) | 27.4 ± 4 | 4.7 ± 0.8 | 2007-2008 |
| Depoorter et al. (2013) | 28 ± 7 | 4.93 ± 1.32 | 2007-2009 |
| Liu et al. (2015) | 26 ± 3 | 6.4 ± 0.7 | 2005-2011 |
| Adusumilli et al. (2013) | 28.3 ± 8.0 | 7.4 ± 2.1 | 1994-2018 |
| | 25.8 ± 8.0 | n/a | 2010-2018 |


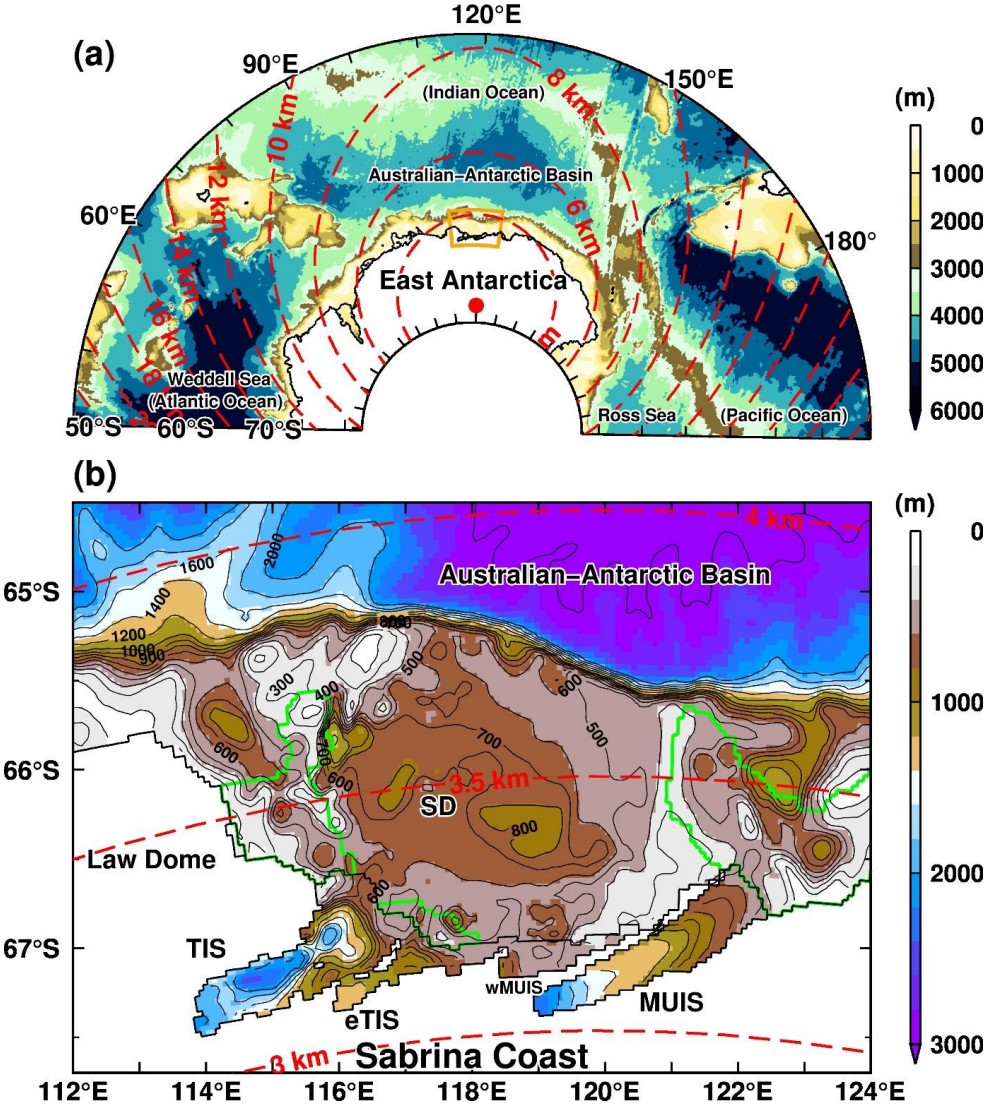

**Figure 1:** Model bathymetry (color) and horizontal resolution (red dashed contours) over (a) East Antarctica and (b) the region off the Sabrina Coast. The red dot in panel (a) represents one of the model's singular points. The orange box in panel (a) shows the region for penal (b). In panel (a), The major place names shown in panel (a) are as follows: the Weddell Sea in the Atlantic Ocean, the Australian Antarctic Basin in the Indian Ocean, and the Ross Sea in the Pacific Ocean. In panel (b), TIS, eTIS, wMUIS, MUIS, and SD represent the Totten Ice Shelf, the eastern Totten Ice Shelf, the western Moscow University Ice Shelf, the Moscow University Ice Shelf, and the Sabrina Depression, respectively. The depth contour intervals are 100 m, 200 m, and 1000 m for regions shallower than 1000 m, 1000-2000 m, and deeper than 2000 m, respectively. The black zigzag lines represents grounding and ice-front lines, and the green line indicates the fastice edge.




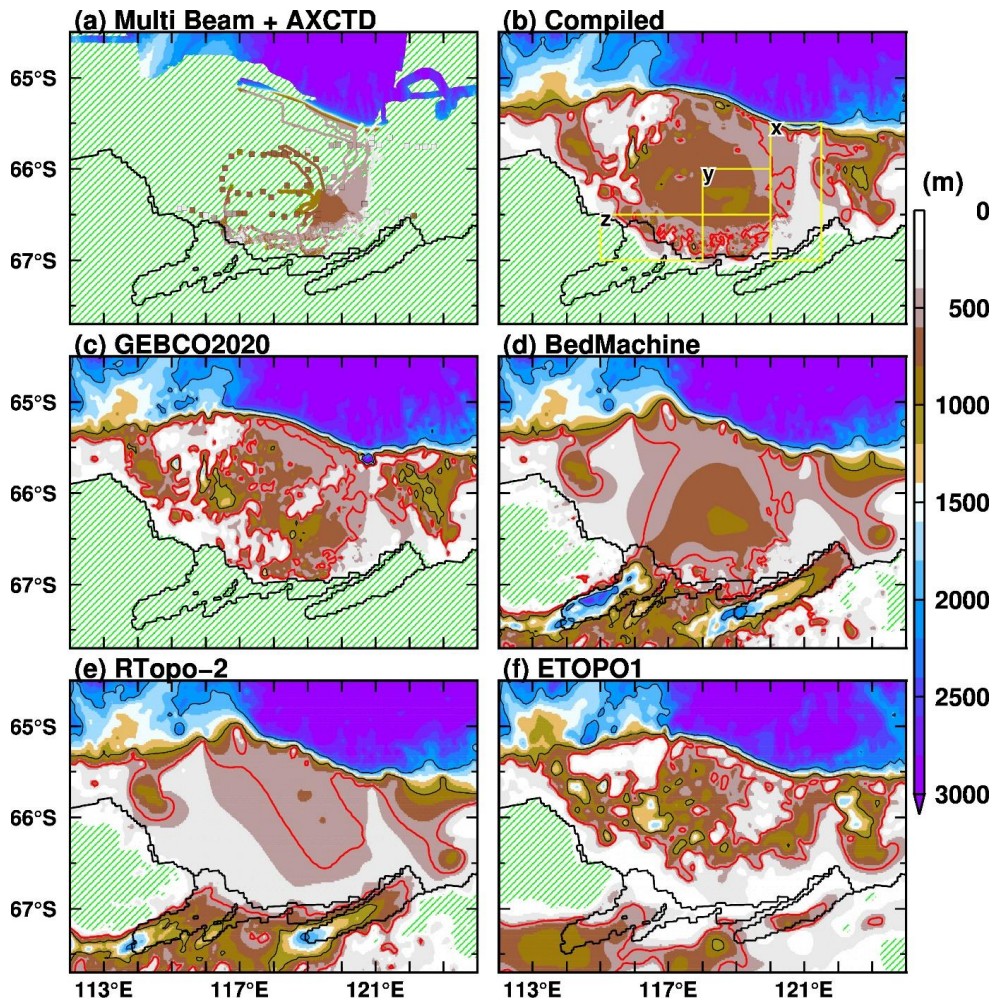

**Figure 2:** Comparison between water depth representation of bathymetric datasets (a: direct observations from multibeam and AXCTD, b: compiled data, c: GEBCO2020, d: BedMachine, e: Rtopo-2, and f: ETOPO1). The black lines in all panels represent the grounding/ice-front line in the model. Squares in panel (a) are the observed depth measurements from AXCTD; yellow rectangles with labels x,y, and z in panel (b) are regions used in comparisons made in the main text. Red contours In panels (b)–(f) show a depth of 500-m to highlight the difference between the bottom topography of the datasets. Green shaded areas represent uncharted areas in panel (a) and land grid points in the other panels.






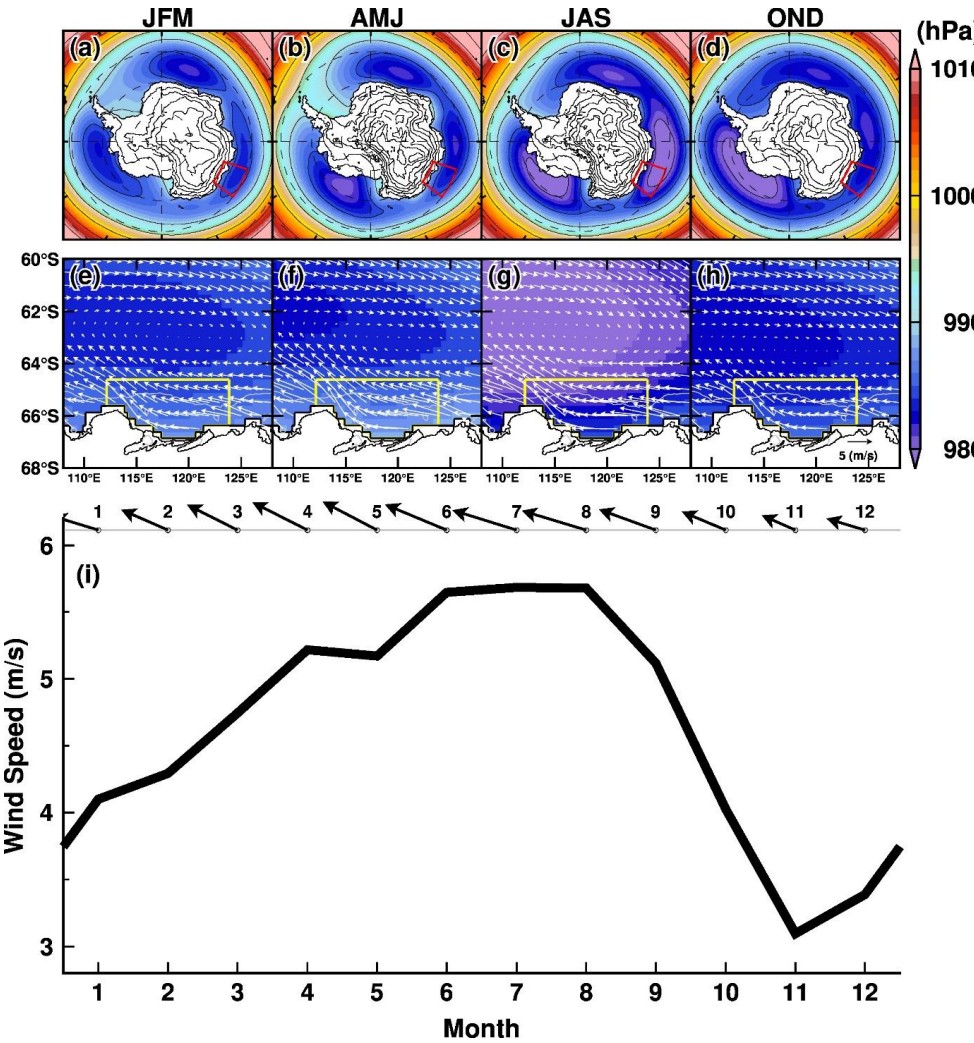

**Figure 3:** Seasonal climatology of atmospheric sea-level pressure (a–d) over the Southern Ocean and (e–f) in the region off Sabrina coast (108°E–128°E). Color and vectors show the 3-month averages of atmospheric sea-level pressure and wind, respectively. The climatology fields were calculated from the ERA5 dataset for the reference period 1981–2010. The red box in panels a–d indicates the region for panels 1025 (e–h); the yellow box in panels (e–h) is the area used to average the 10-m wind for panel (i). In panel (i), the monthly climatologies of wind vectors and wind speeds are shown.






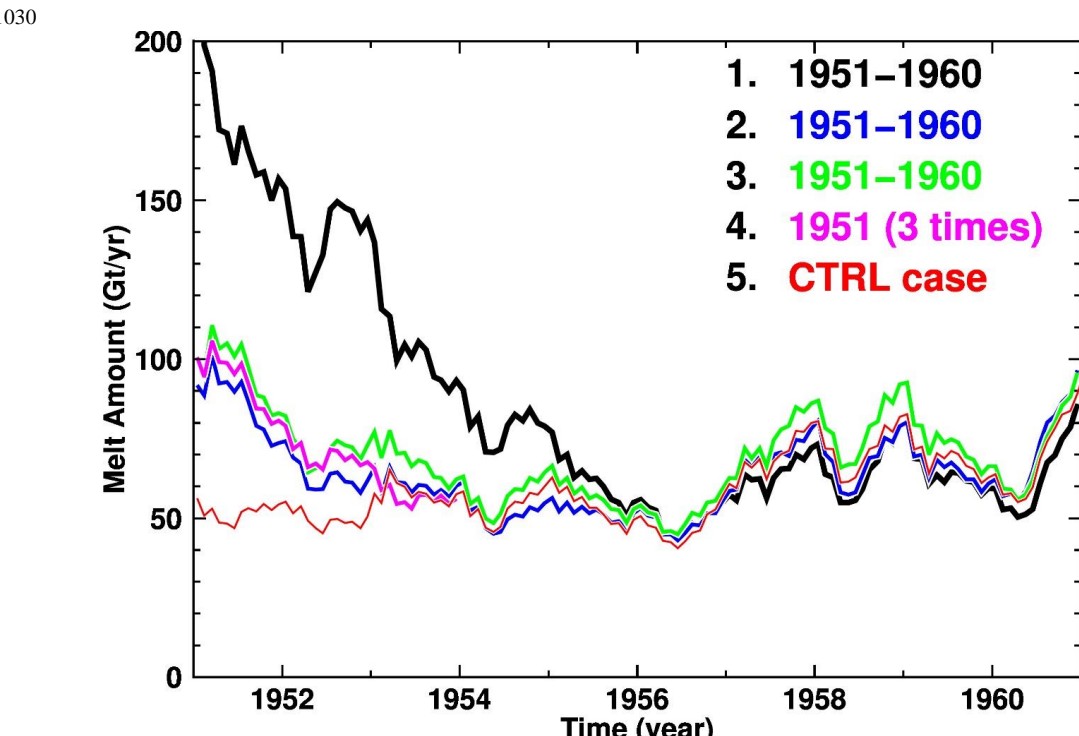

**Figure 4:** Time series of basal melt amount at ice shelves along the Sabrina Coast in model spin-up stages. After three cycles of 10-year integration with the 1951–1960 forcing (black, blue, and green), a three-year integration repeatedly with 1951 forcing (pink) was conducted to obtain the initial condition for CTRL case (red).


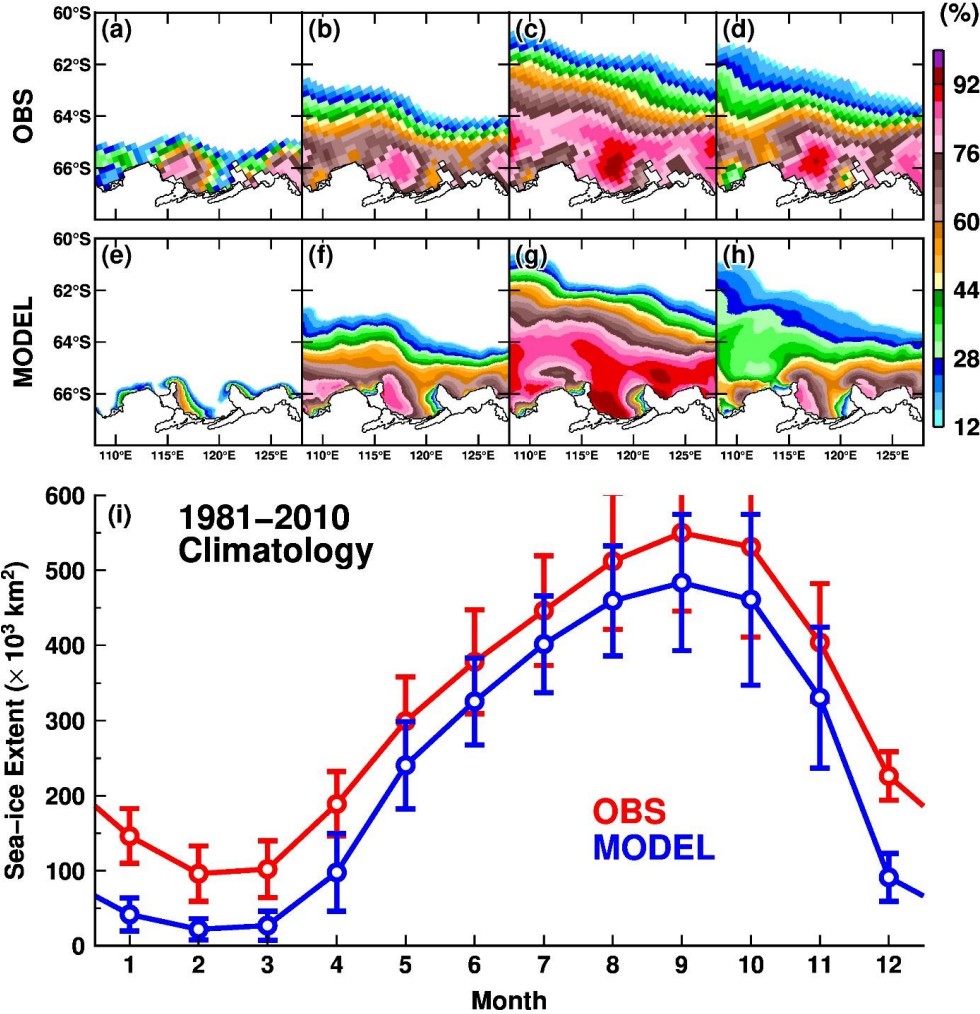

**Figure 5**: Seasonal cycle of observed and modeled sea-ice climatology averaged over the reference period (1981–2010). (a–h)
Maps of sea-ice concentration in a longitudinal range from 108°E to 128°E (a–d: observation and e–h: model) and monthly
sea-ice extent (red: observation and blue: model) within the longitudinal band.



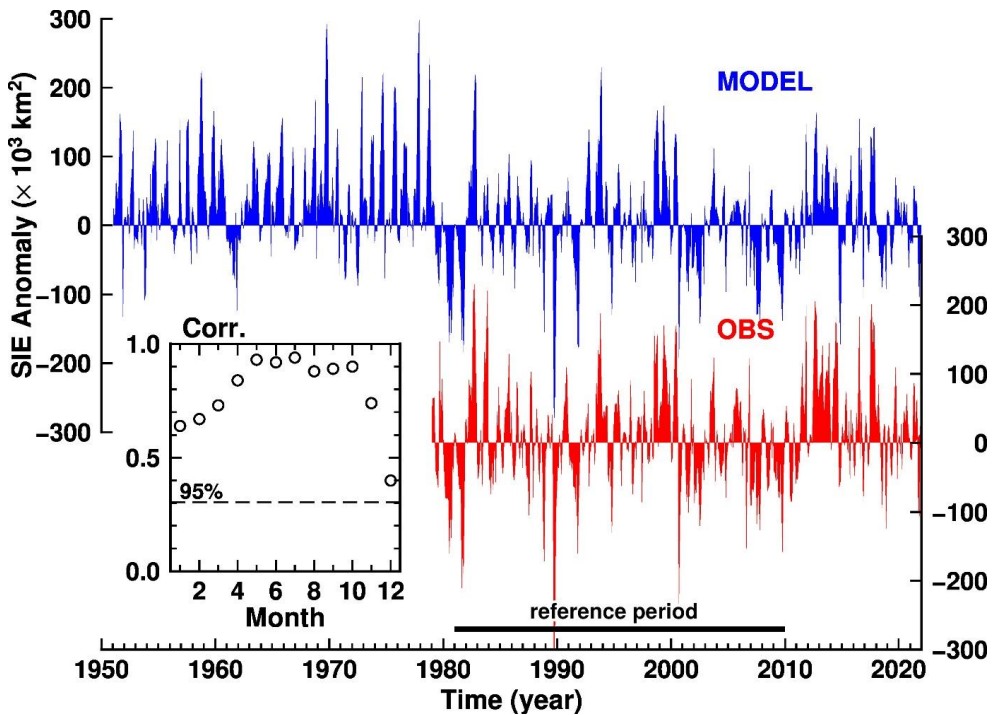

**Figure 6:** Time series of monthly sea-ice extent anomalies (blue: model for the period 1951–2021 and red: observation for the period 1979–2021). The anomalies are deviations of the regional monthly sea-ice extent from the monthly climatologies (Fig. 5i). Bottom-left small panel shows monthly correlation coefficients of sea-ice extent anomalies for the period 1979–2021 between the observation and model. The dashed line indicates the 95% significant level.






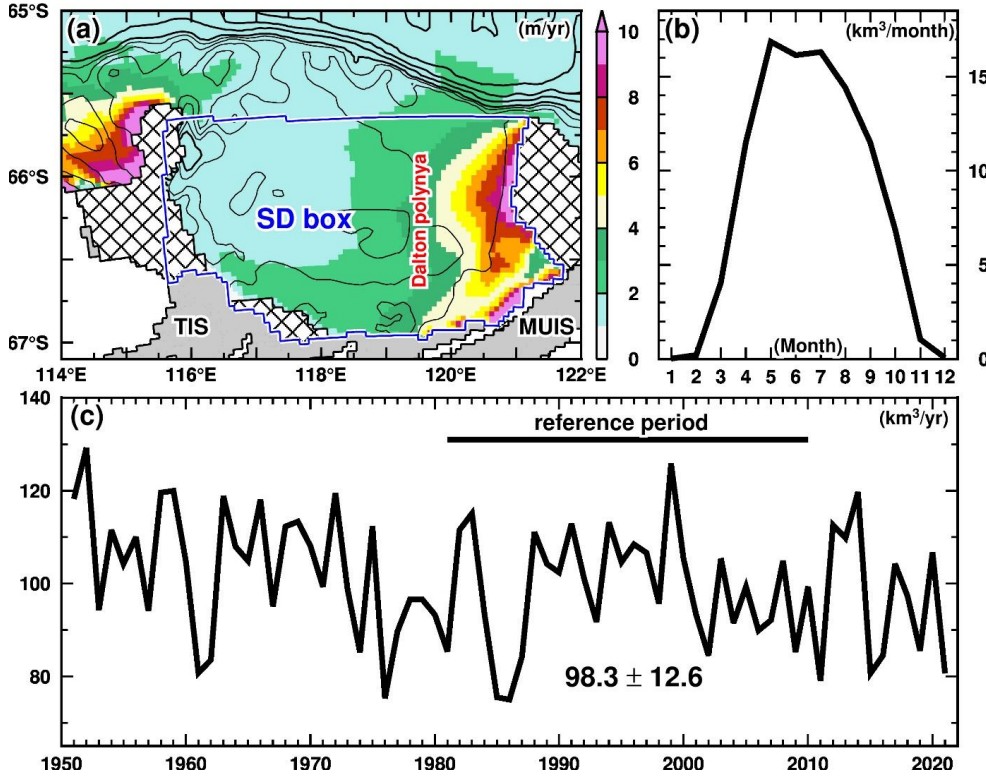

**Figure 7:** Modeled sea-ice production. (a) Annual sea-ice production map, (b) monthly sea-ice production over the SD box, and (c) the interannual variability. The annual and monthly sea-ice production in panels a and b are obtained from the climatology of the reference period (1981–2010).




**Figure 8:** Maps of (a) annual mean ice-shelf basal melt rate, (b) seasonal amplitude, and (c–d) months of the maximum and minimum. Climatology averaged over the reference period 1981–2010 is used for the plots.




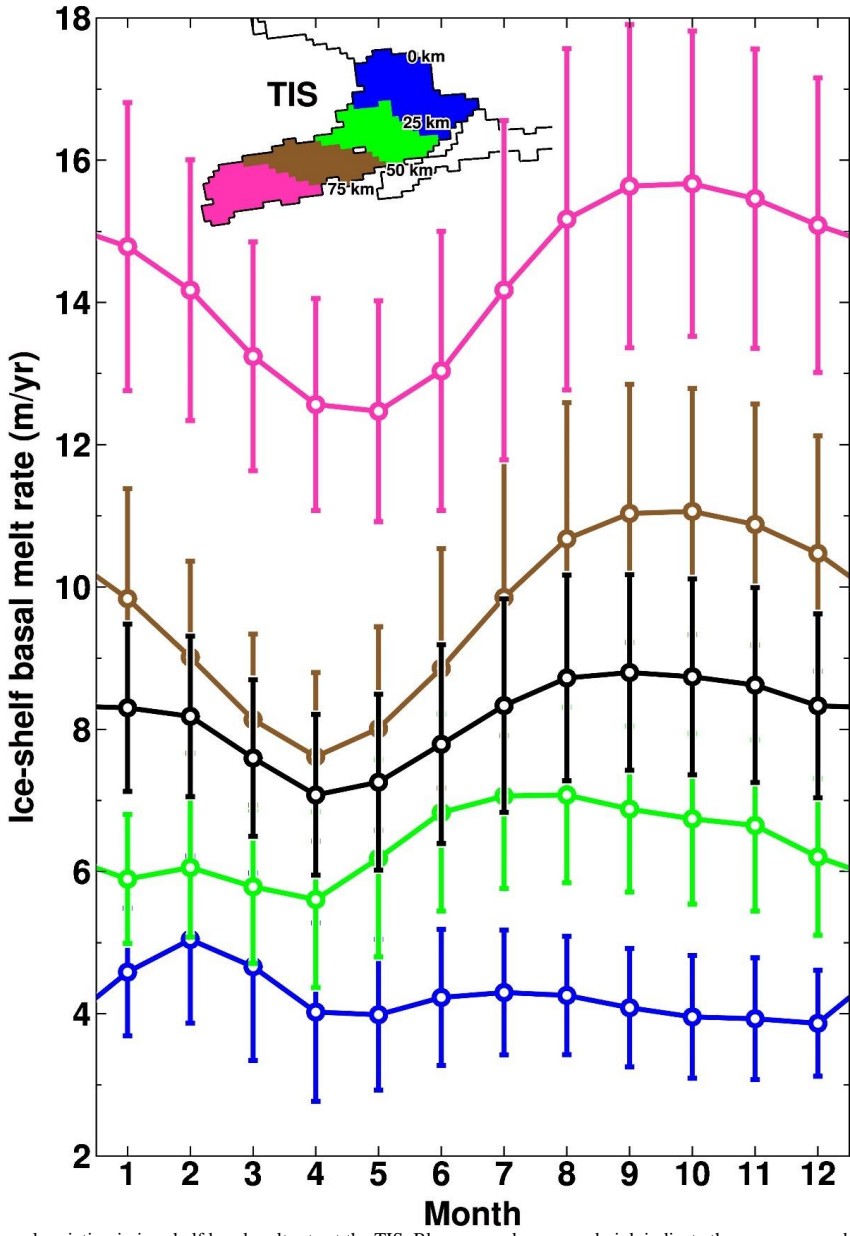

**Figure 9:** Seasonal variation in ice-shelf basal melt rate at the TIS. Blue, green, brown, and pink indicate the area-averaged melt rate in the four regions shown in the inset. Black shows the mean melt rate averaged over the TIS; circles and error bars show the mean and standard deviation of the melt rate for the reference period (1981–2010).






**Figure 10:** Time series of ice-shelf basal melt amount at four ice shelves along the Sabrina Coast (a: TIS, b:eTIS, c:wMUIS, and d:MUIS) for the period 1951–2021. Blue and red lines show monthly and annual basal melt amounts, respectively; gray and black lines show results

from the CKDRF case.




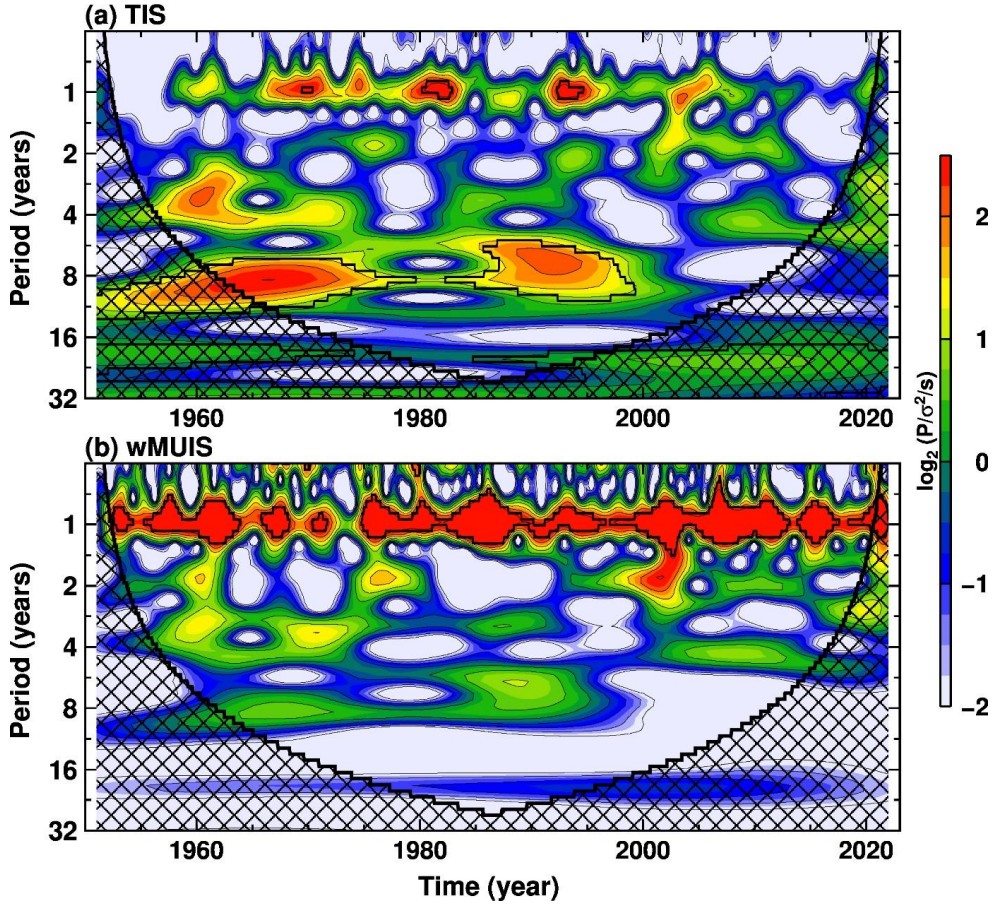

**Figure 11:** Wavelet power spectra for the ice-shelf basal melt rate in the TIS and wMUIS. The power spectrum is normalized by the variance ($\sigma^2$) and the scale (s). The vertical axis is the period (in year), and the horizontal axis is time (year). The regions enclosed by thick contours indicate time-period zones greater than the 95% confidence level. Cross-hatched regions indicate the cone of influence, where edge effects are not negligible.

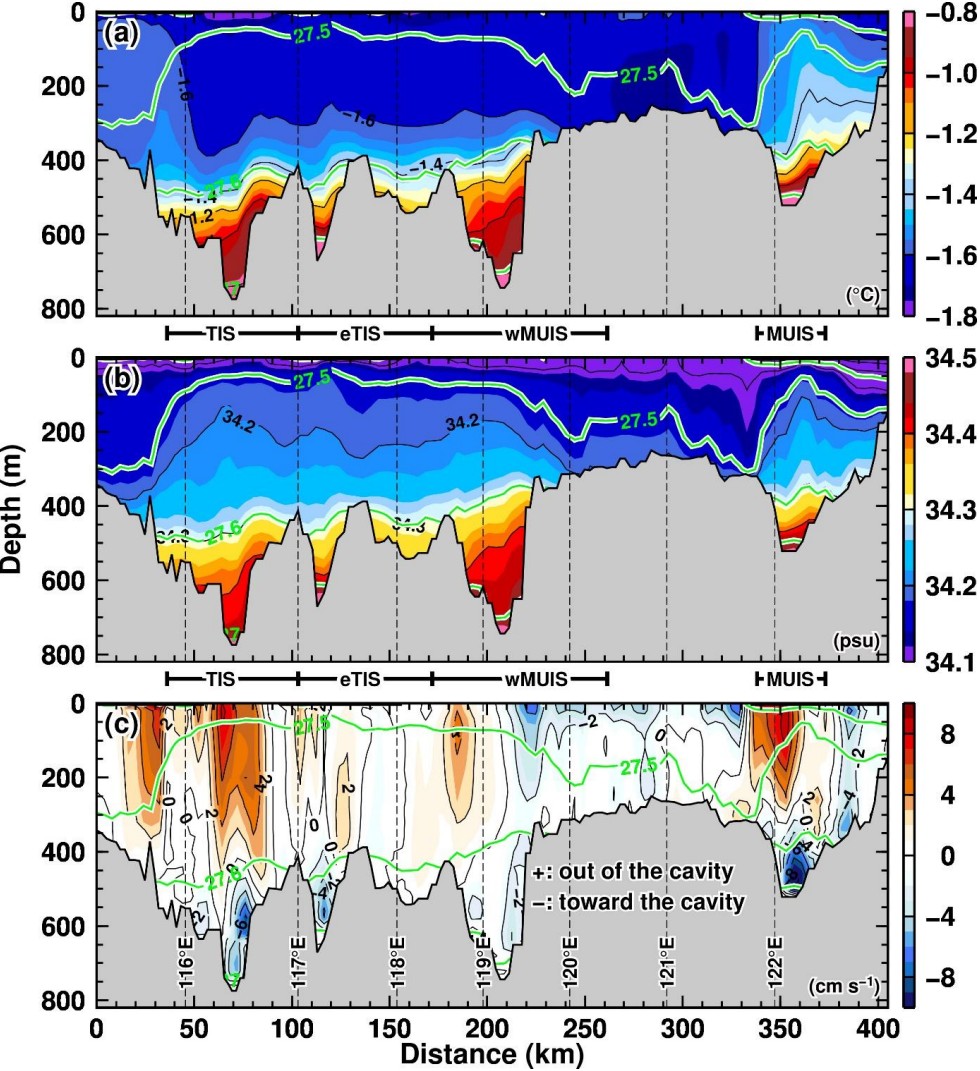

**Figure 12:** Vertical profiles of (a) potential temperature, (b) salinity, and (c) velocity along a 5-km off along a line section of the Sabrina Coast. The annual-mean variables averaged over the reference period (1981–2019) are used for the plots. Green contours indicate the potential density anomaly. Positive (negative) velocity indicates the ocean flow directed offshore (coast/ice shelf).


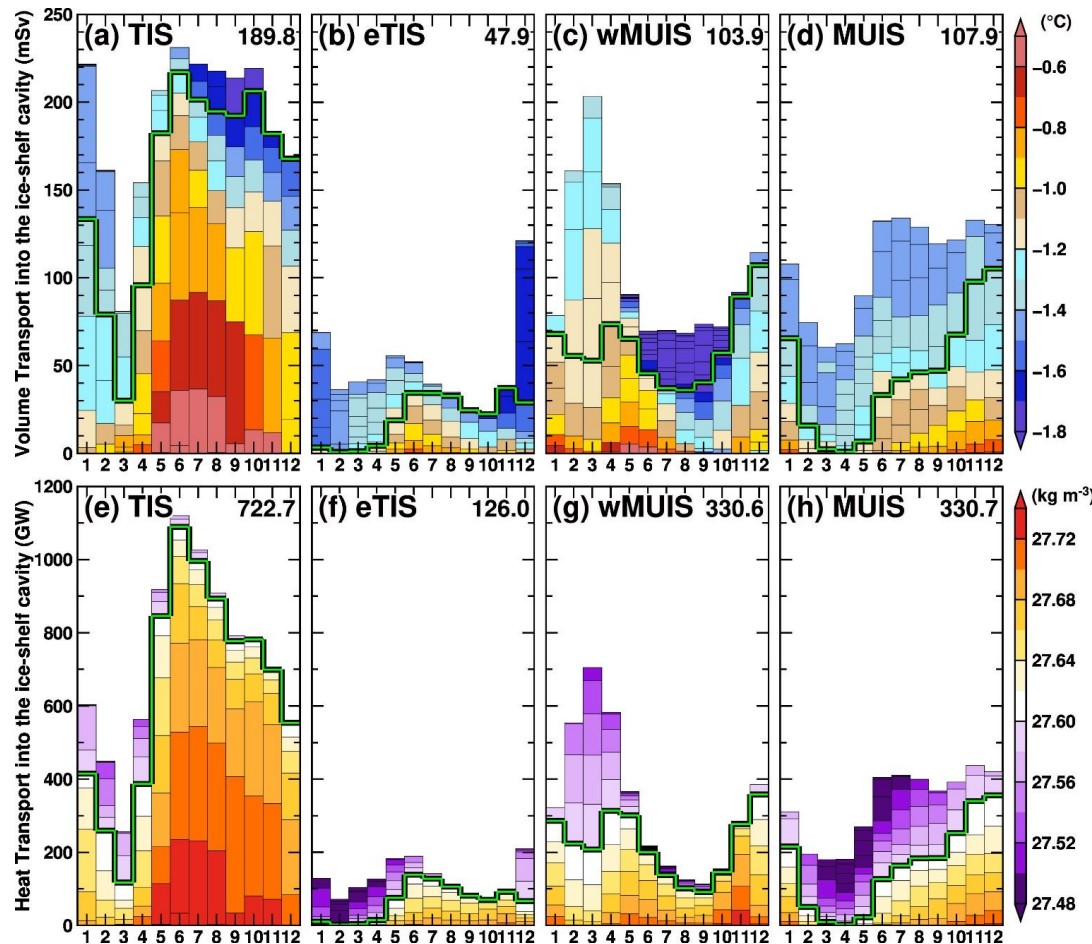

**Figure 13:** Seasonal variations in the net volume and heat transports into the four ice-shelf cavities across the ice-shelf front (TIS, eTIS, wMUIS, and MUIS from left to right). Colors in the upper (a–d) and lower (e–h) panels indicate potential temperature and density in 0.02 kg m$^{-3}$ potential density bins, respectively. Green line shows the density interface of 27.60 kg m$^{-3}$. All variables are averaged over the reference period (1981–2010).



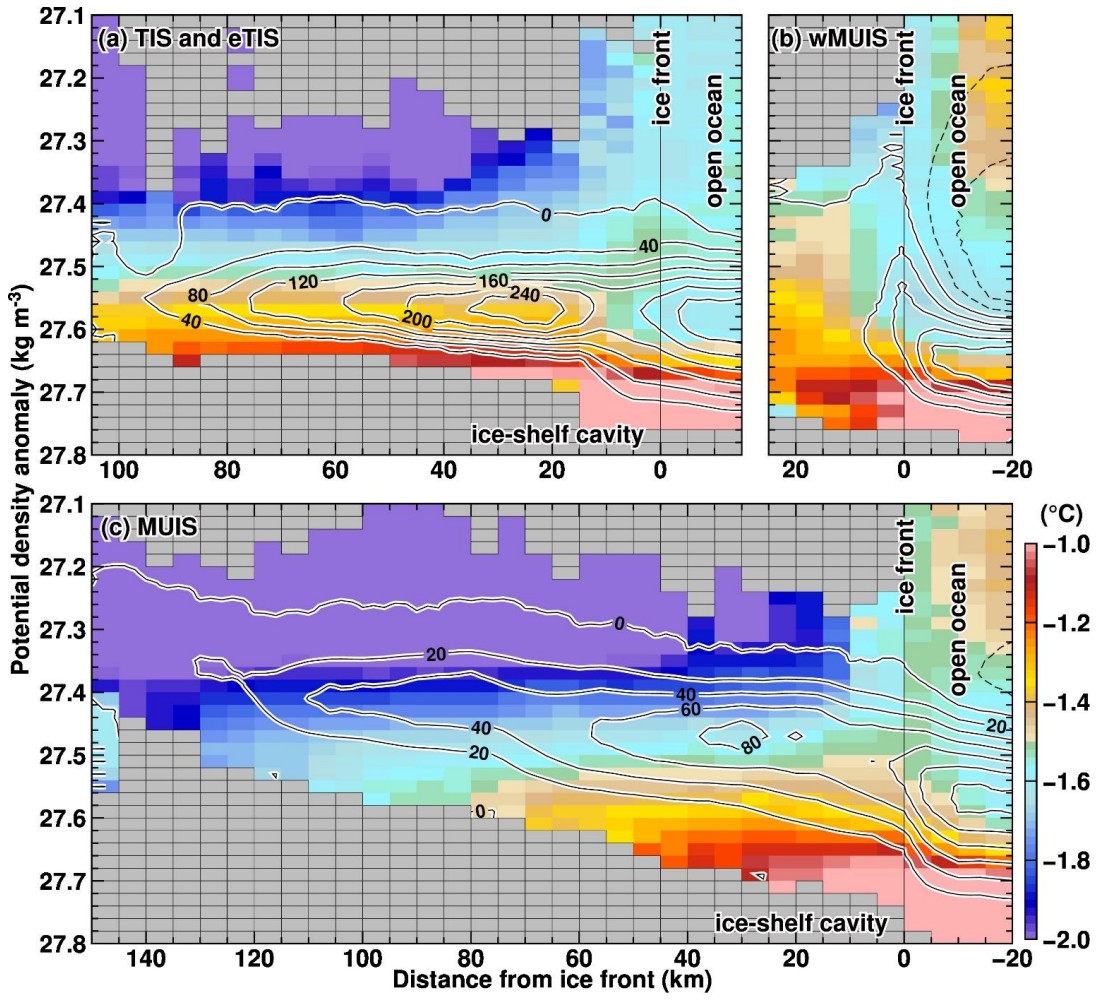


**Figure 14:** Vertical overturning circulations under (a) the TIS-eTIS, (b) wMUIS, and (c) MUIS cavities. Contours and colors show overturning stream function and ocean temperature in the distance-density space. The horizontal axis is the distance from the ice-shelf fronts, and the vertical axis is the potential density anomaly. The results are the annual-mean climatology averaged over the reference period (1981–2010).


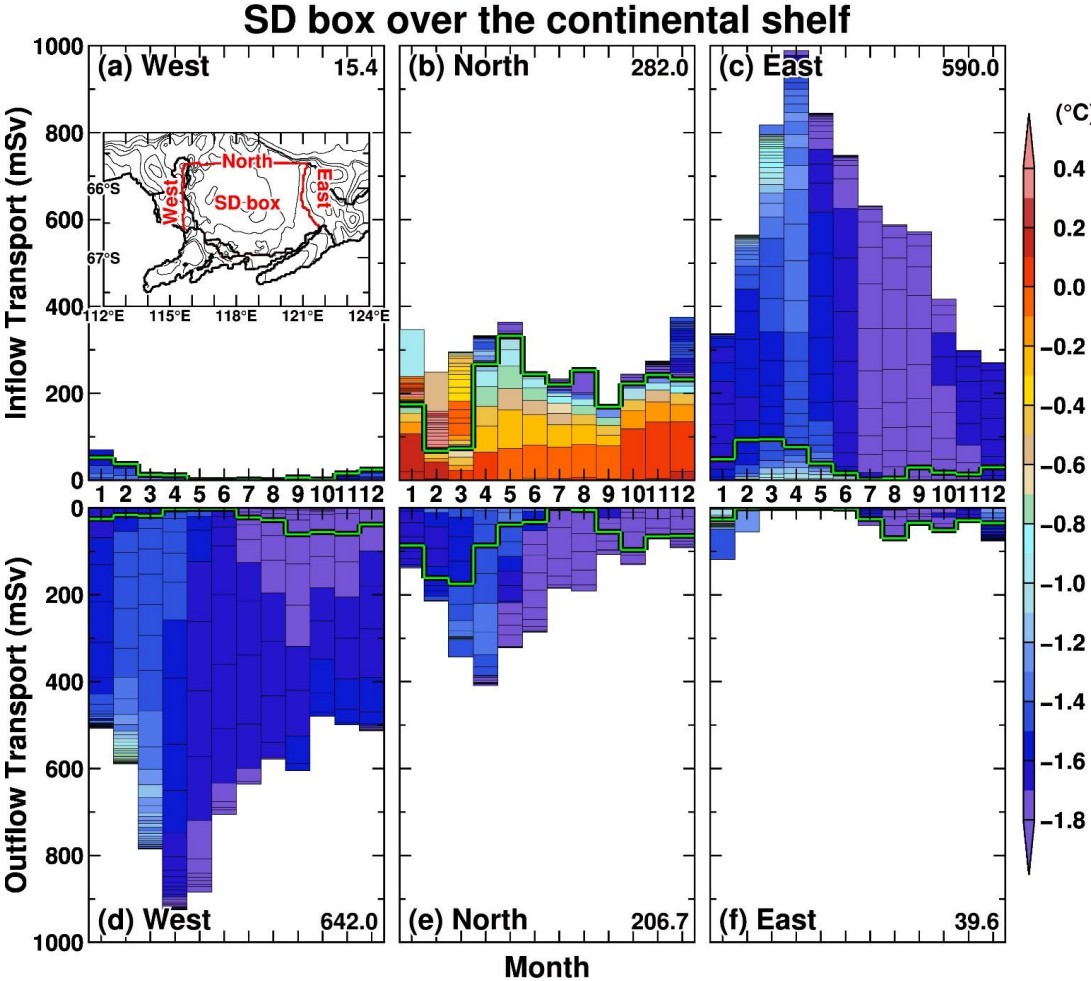

**Figure 15:** Seasonal variations in inflow and outflow volume transports across the boundaries of the SD box (left: western
boundary, middle: northern boundary, and right: eastern boundary). Colors in the panels indicate the potential temperature in
0.02 kg m⁻³ density bins. Green line shows the density interface of 27.60 kg m⁻³. All variables are averaged over the reference
period (1981–2019). The number on the right side of each panel indicates the annual-mean volume transport.




**Figure 16:** Horizontal distributions of (a) annual-mean ocean temperature averaged over depths below 400m and (b) the peak month. These maps are based on the annual/monthly climatology for the reference period 1981–2010. Boxes labeled with A, B, C, and D show areas used for the vertical profiles in Fig. 17.






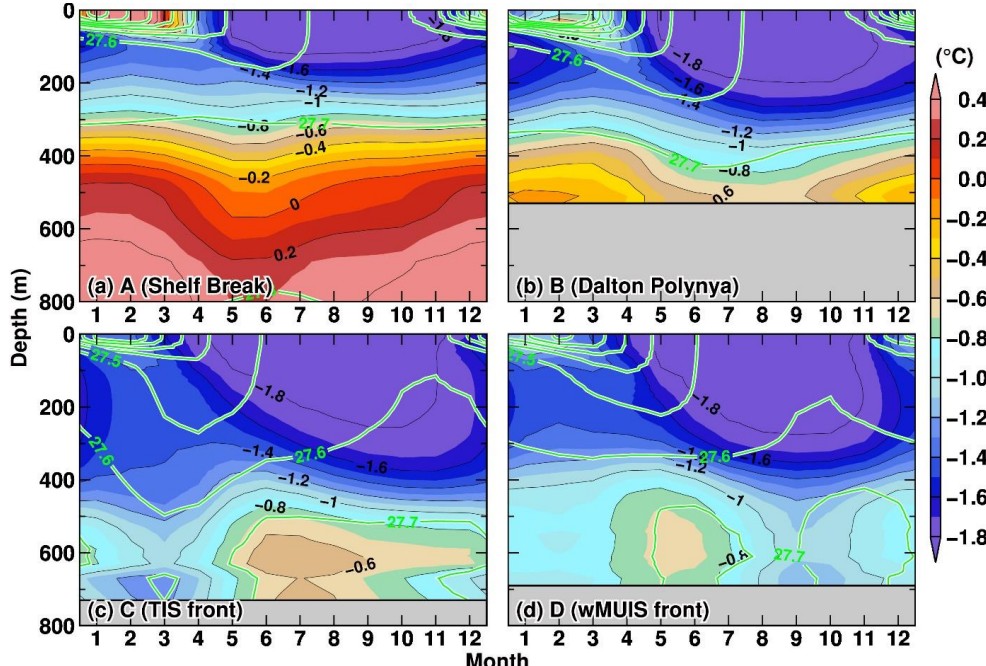

**Figure 17:** Seasonal variation in potential temperature (color) and potential density anomaly (green contours) average over the boxes in Fig. 16. The panels are based on the monthly climatology for the period 1981–2010.





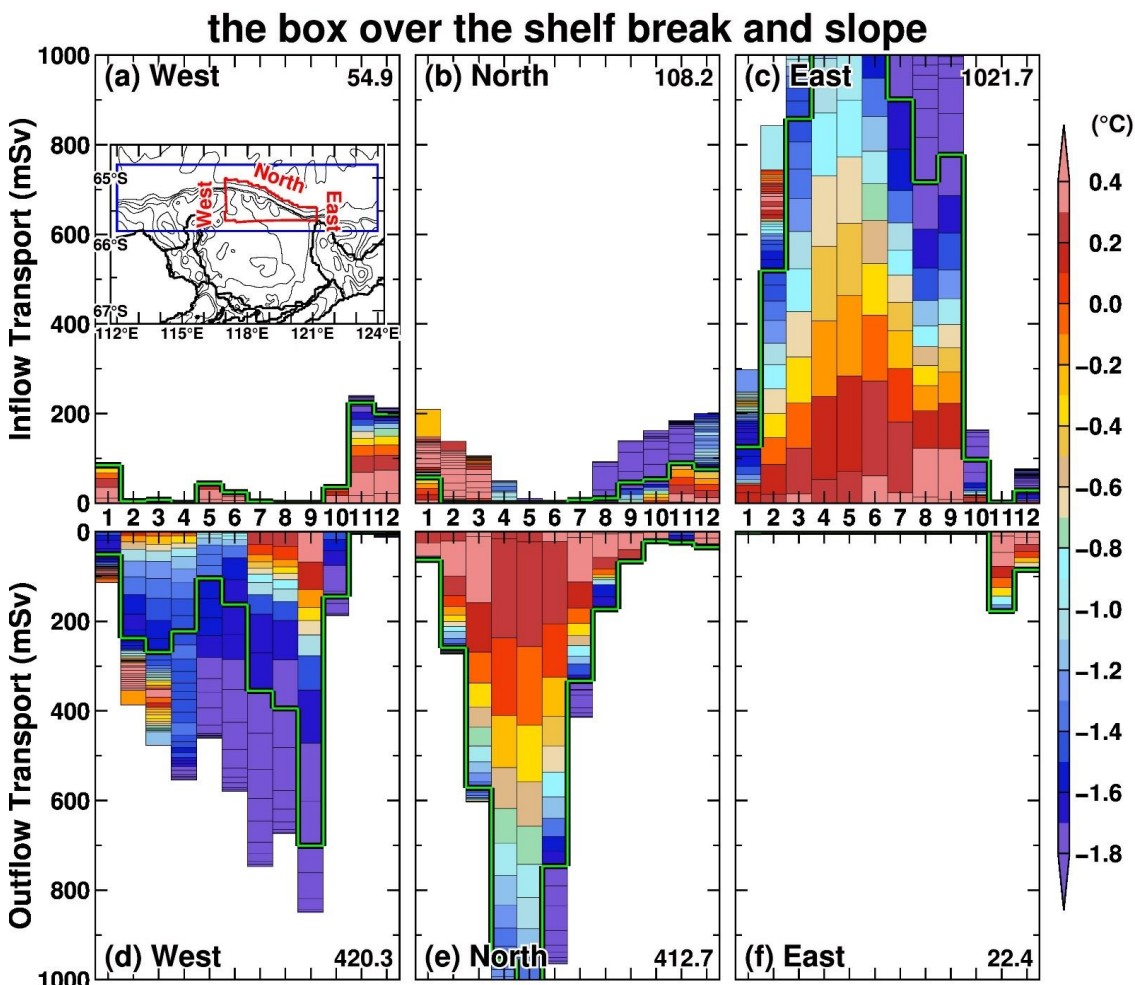

**Figure 18:** As in Fig.15 but for the Slope Box. A blue box in the inset shows an area for Figs. 19 and 20.




**Figure 19:** Maps of ocean flows in the depth range from the surface to 100 m in (a) December, (b) June, and (c) September. The vector shows the ocean flow direction with color showing the magnitude. Thick black curves show 500 and 2000-m depth contours, and gray curves show the depth contour with 200-m intervals. Magenta line indicates the boundaries of the Slope and SD boxes.






**Figure 20:** As in Fig. 19, but for a depth range from 400 to 600 m.





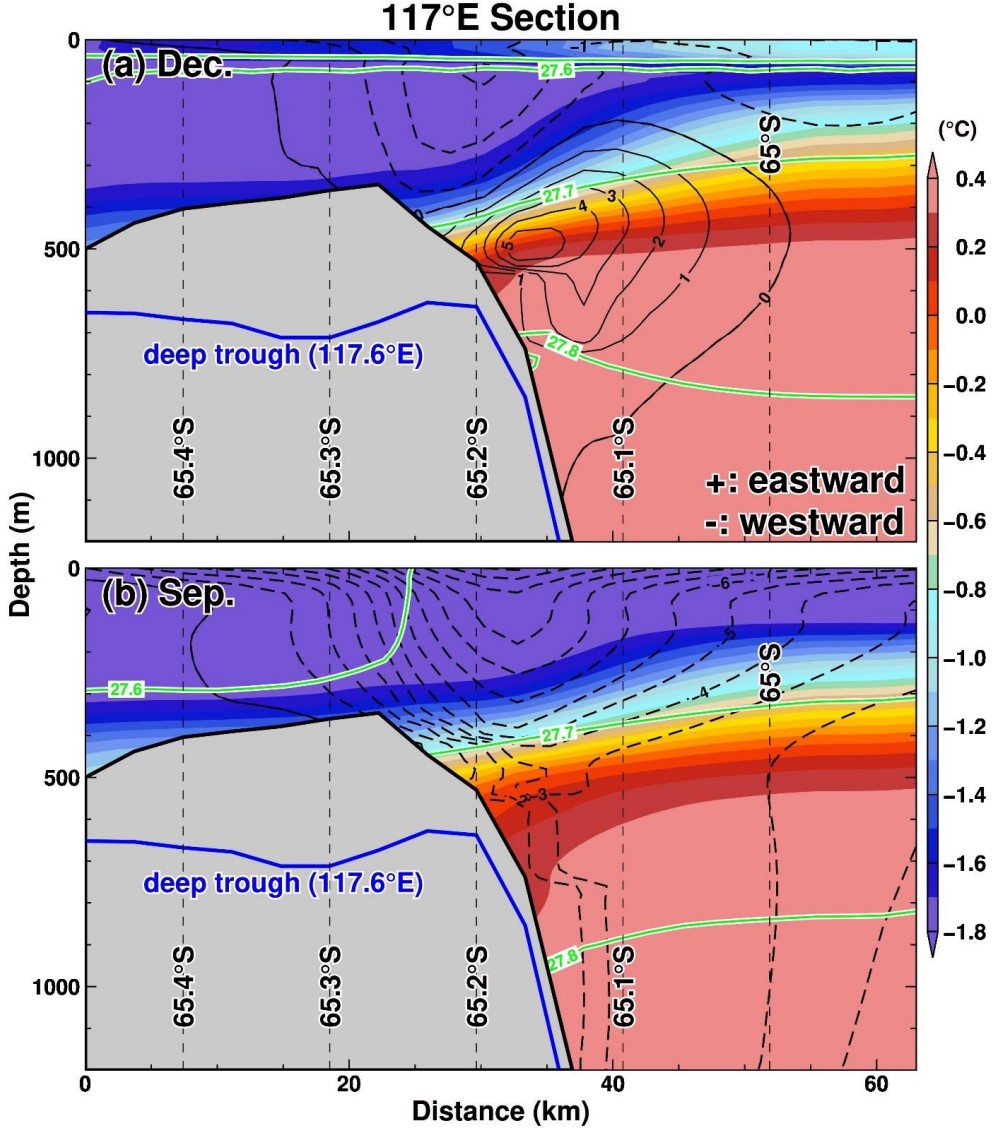

**Figure 21:** Vertical profiles of potential temperature (color), potential density anomaly (green contours), and east-west velocity (black contours) along 117°E in (a) December and (b) September. The blue line shows the bottom topography along the 117.6°E section. Positive and negative values of the velocity indicate eastward and westward, respectively. The variables averaged over the reference period 1981–2010 are used for the plots.

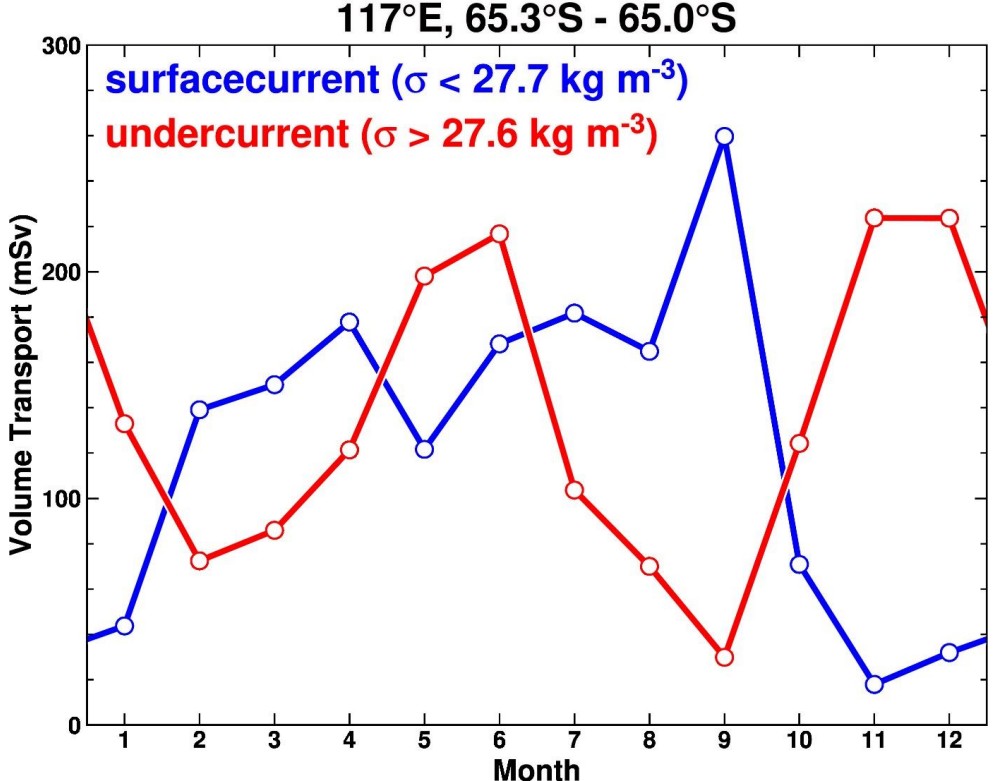

**Figure 22:** Seasonal variation in volume transports of the westward-flowing surface current (blue) and the eastward-flowing
undercurrent (red). The monthly climatology averaged over the reference period 1981–2010 is used for the plot.





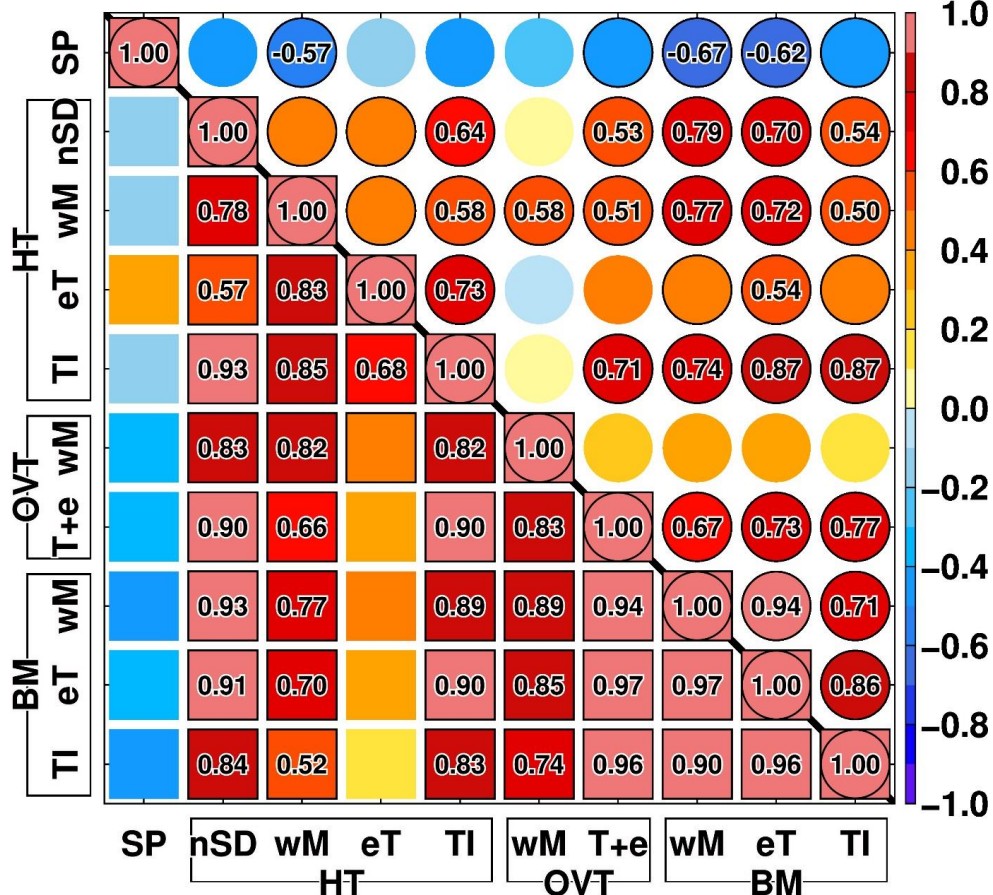


**Figure 23:** Correlation coefficients between two variables. The variables are sea-ice production (SP), heat transport (HT), overturning circulation (OVT), and ice-shelf basal melting (BM). The short abbreviations of nSD, wM, eT, TI, T+e stand for the northern boundary of the SD box, western MUIS, eastern TIS, and TIS+eTIS, respectively. Squares show correlation coefficients after 7-yr running mean data, and circles show correlation coefficients after removing the running mean. Marks

enclosed with black lines indicate exceeding 90% confidence level (0.43), and numbers indicate exceeding 95% confidence level (0.50). The confidence levels are estimated from the Monte Carlo methods (generating random samples, taking 7-yr running mean, estimating the length of autocorrelation, and calculating effective degrees of freedom). The same confidence levels are used for the high-frequency year-to-year areas (circles).






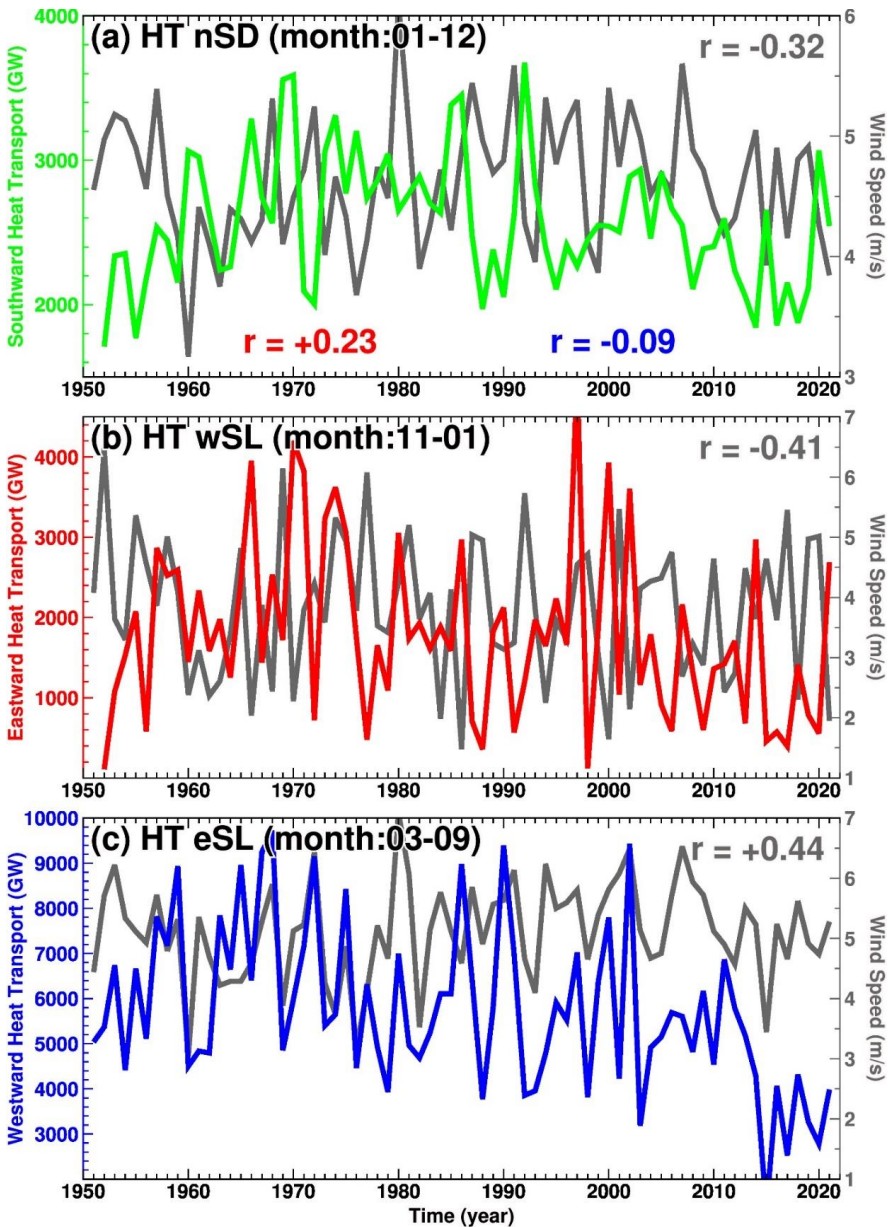

**Figure 24:** Time series of ocean heat transports and wind speed. Colored lines in panels (a), (b), and (c) are the southward component across the SD Box's northern boundary averaged over the year, the eastward components across the Slope Box's western boundary averaged from November to January, and the westward components across the Slope Box's eastern boundary averaged from March to September, respectively. The correlation with the coastal wind speed is shown in the upper right corner. In panel (a), correlations of the annual-mean heat transport with the eastward and westward heat transports are shown in red and blue, respectively.