# Peer review of "Modeling seasonal-to-decadal ocean-cryosphere interactions along the Sabrina Coast, East Antarctica"

_The Cryosphere, 2023_

## Author Response (AR1)

**Response to the editor's comments**

(NB *italicized text in box* is comments from the editor. Some numbers with bold fonts were inserted for our convenience in addressing the comments.)

*Reviewer 1 Comment (1) L165: the velocity-independent scheme*

**(1)** - *I interpreted the reviewer's comment as a request for a more in-depth discussion on how the velocity-independent scheme might impact the results, so it would be good to see some more details about this. For example, in the Methods section, are there studies that show key differences between melt rates predicted using velocity-dependent vs velocity-independent schemes? What might we expect in terms of the differences? Where might we expect the velocity-independent scheme to be less robust? Do we expect that the model grid configuration used in this study is not optimal for the velocity-dependent scheme? Does this indicate that the model is too coarse resolution? And in the Discussion section, how does what we know about the velocity-dependent vs -independent schemes compare with the results here? Do we have more or less confidence in the modelled melt rates in some of the regions of the Totten Ice Shelf?*

**(2)** - *The proposed amended text for the Methods includes "...strongly influenced by both horizontal and vertical grid resolutions...". Do we expect that the velocity-independent scheme has no dependence on grid resolution?*

**(1) and (2)** The parameterization of ice-ocean interactions at ice-shelf base is an active research topic from both observational and modeling perspectives, given its importance in determining basal melt rate and amount. The basal melt rate and amount estimated in our model are the outcome of a complex combination of factors, including ice-shelf cavity topography (bathymetry and ice-shelf draft), model resolution, represented physical processes, and the selected parameterizations. While the total basal melt amount in our model is reasonably consistent with values derived from satellite observations (Table 2), we admit that our results are not perfect due to uncertainties related to bathymetry, parameterization, and the representation of physical processes (e.g., tide and submeso-scale eddy).

Upon discretization of the equations, numerical models have inherent limitations in their ability to directly represent phenomena occurring at scales smaller than the model's grid size. Additionally, the choice of discretization introduces its own set of strengths and weaknesses, leading to uncertainties. For example, models using z-coordinates are known to underestimate downslope flow of dense water along continental slopes. Similarly, these models with an ice-shelf component could potentially underestimate buoyant upslope flow along the ice-shelf bases. On the other hand, terrain-following coordinate systems can more naturally represent flows along the slopes but introduce errors related to pressure gradients and even require smoothing to deal with steep topography.

Modeling ice-shelf basal melt rates involves capturing processes at different scales, from large-scale ocean circulation to localized melting phenomena. In our study, we aimed to resolve relatively large-scale ocean circulation, focusing primarily on the inflow of warm water from the Sabrina Depression to the ice-shelf cavities. We consider velocity-dependent parameterizations to be more useful for higher-resolution models targeting smaller scales. It's worth noting that even within velocity-dependent parameterizations, there are parameters to be made in the formulation. We opted for velocity-independent parameterizations, as our model development considered these resolution-dependent uncertainties. We chose to use observational-based

parameterizations as opposed to those that z-coordinate models tend to underestimate. This was a deliberate choice on our part.

In the revised manuscript, we have expanded the explanation of using a velocity-independent scheme in the Methods section. Specifically, we briefly outline potential differences in the ice-shelf melting and processes that could arise when using velocity-dependent and velocity-independent schemes. We also clarify that the thermal forcing due to warm water inflow, which is the primary driving forcing for ice-shelf basal melting and is the focus of our study, is adequately captured in our model setup. In the Discussion section, we cited the previous modeling studies comparing the two types of parameterizations (Mueller et al., 2012; Dansereau et al., 2014) and speculated how our melt patterns might differ if a velocity-dependent scheme were employed.

Thank you once again for your valuable comments. We believe these amendments have helped us respond more effectively to the reviewer's comment.

**L168–180**

We used observation-based coefficients of the thermal and salinity exchange velocities for the ice shelf ($\gamma_t$=1.0×10$^{-4}$ ms$^{-1}$, $\gamma_s$=5.05×10$^{-7}$ ms$^{-1}$, Hellmer and Olbers, 1989), and applied one-tenth coefficients for landfast ice to consider the difference in the tidal speed between the ice-shelf cavity and the open ocean in the parameterization. In our previous ocean modeling for Lützow-Holm Bay, we confirmed that the magnitude of landfast-ice melting did not significantly affect the variability of the inflow of mCDW onto the continental shelf (Kusahara et al., 2021). Regarding the ice-ocean parameterization, it is worth noting that many ice shelf-ocean modeling studies utilized velocity-dependent coefficients for solving ice-ocean interactions (Holland and Jenkins, 1999; Malyarenko et al., 2020). Given that the magnitude of ocean velocity under ice shelves can be strongly influenced by various factors including both horizontal and vertical grid resolutions (Gwyther et al., 2020), as well as unmodeled processes like tidal effects and sub-mesoscale eddy activities, we opted for the velocity-independent scheme of Hellmer and Olbers (1989) that assumed a constant ocean velocity 15 cm/s. It should be noted that using the velocity-independent parameterization also implies that we ignore the local enhancement in melt rates due to the formation of buoyant upslope plumes arising from the upstream ice-shelf melting. However, such effects are considered secondary compared to the primary influence of the dominant heat supply through warm water inflow into the ice-shelf cavities.

**L620–629**

For instance, the accuracy of the bathymetric data underneath ice shelves can greatly affect the representation of circulation under the ice shelf. The ocean velocity at the ice-shelf bases has an impact on both the patterns and magnitude of ice-shelf basal melting. This effect is especially pronounced when a velocity-dependent parameterization is used for thermal and salinity exchanges at the ice-ocean interface (Mueller et al., 2012; Dansereau et al., 2014). It's worth noting, however, that this study employed a velocity-independent parameterization. While this approach captures the ice-shelf basal melting driven primarily by the dominant thermal forcing originating from the mCDW intrusion, it cannot account for secondary effects like enhanced melting patterns due to buoyancy-driven upslope flows generated by the upstream ice-shelf meltwater. In scenarios where velocity-dependent parameterization is used, the melting pattern in the model might change in regions with stronger currents, offering a potentially more intricate view of the ice-ocean interactions. Incorporating the updated datasets,…

Reviewer 2

**(3)** - *Comment (8). It would be good to clearly state that the difference between the inflow and outflow transport is accumulated only when the difference is positive (i.e. there is net inflow).*

**(4)** - *Comment (9). In the updated red sentence, please refer to the corresponding figure 15*

**(3)** We have added a sentence.

**L370–373**

The volume transports and the mean temperatures were calculated for each 0.02 kg m$^{-3}$ bin of the potential density for the inflow and outflow components. Net inflow transport was determined only when the difference between inflow and outflow transport was positive. The upper panels in Fig. 7 show the net inflow volume transports in each density bin and the mean temperatures of the inflow component.

**(4)** We have referred to the figure. Thank you for your careful reading.

**Response to the comments from Dr. Chengyan Liu (Reviewer #1)**

(NB *italicized text in box* is comments from the reviewer. Some numbers with bold fonts were inserted for our convenience in addressing the comments.)

*General comments:*

*This paper presents an investigation of the ocean-cryosphere interactions off the Sabrina Coast of Wilkes Land, East Antarctica. Based on a coupled ocean–sea ice–ice shelf model, the authors studied the sea ice evolution, the basal melting of ice shelves, the properties of water masses and circulations, modified Circumpolar Deep Water (mCDW) intrusions over the shelf, the oceanic heat and volume transports, and the meridional overturning circulations within the sub-ice-shelf cavity around the Sabrina Coast.*

*The state-of-the-art topography data over the Sabrina Coast have been constructed and introduced in the coupled model, and the overturning ocean circulations within the sub-ice-shelf cavities are also shown in this study for the first time. The mechanism responsible for the differences in temporal variability between the basal melting of the Totten Ice Shelf and Moscow University Ice Shelf has been discussed, and the authors found that both mCDW intrusions and sea ice production contribute to the regional differences between the two sub-ice-shelf cavities. More interestingly, the model has captured an eastward undercurrent over the continental slope, which may significantly regulate the simulated seasonal variabilities of onshore heat transport.*

*It is very topical because ocean-cryosphere interactions around the Sabrina Coast are key processes for the marine ice sheet instability around East Antarctica, which has global implications for climate change and the sea level rising. I believe that this manuscript is very interesting to the Antarctic science community. My comments are given below, and I recommend the manuscript for publication in TC after minor revision.*

Thank you very much for your careful reading of our manuscript and your constructive comments. We are pleased to hear that you find our work interesting, and that you recommend publication in TC. Your feedback has been invaluable for improving the quality of our paper, and we are committed to addressing your comments and suggestions.

*Specific Comments:*

**(1)** *L165: 'We used observation-based coefficients of the thermal and salinity exchange velocities for the ice shelf (γt=1.0×10−4 ms−1, γs=5.05×10−7 ms−1, Hellmer and Olbers, 1989)'*

*A fixed frictional velocity has been employed by the model. It would be nice if the authors could make a few discussions about the potential discrepancy of the fixed frictional velocity for the thermal and salinity exchanges at the ocean-ice shelf interface, by comparing it to the parameterization of the velocity-dependent scheme.*

**(1)** Thank you for your comment on the velocity-independent ice-ocean parameterization used in our model. We appreciate the suggestion to discuss the potential impact of the choice of parameterization. In the revised manuscript, we have added sentences about why we opted for a velocity-independent parameterization in the Method section. Additionally, in the Discussion section, we have included sentences on how a velocity-dependent scheme could alter the model's ice-shelf melting, citing literature.

**L168–180 (in Method section)**

We used observation-based coefficients of the thermal and salinity exchange velocities for the ice shelf ($\gamma_t$=1.0×10$^{-4}$ ms$^{-1}$, $\gamma_s$=5.05×10$^{-7}$ ms$^{-1}$, Hellmer and Olbers, 1989), and applied one-tenth coefficients for landfast ice to consider the difference in the tidal speed between the ice-shelf cavity and the open ocean in the parameterization. In our previous ocean modeling for Lützow-Holm Bay, we confirmed that the magnitude of landfast-ice melting did not significantly affect the variability of the inflow of mCDW onto the continental shelf (Kusahara et al., 2021). Regarding the ice-ocean parameterization, it is worth noting that many ice shelf-ocean modeling studies utilized velocity-dependent coefficients for solving ice-ocean interactions (Holland and Jenkins, 1999; Malyarenko et al., 2020). Given that the magnitude of ocean velocity under ice shelves can be strongly influenced by various factors including both horizontal and vertical grid resolutions (Gwyther et al., 2020), as well as unmodeled processes like tidal effects and sub-mesoscale eddy activities, we opted for the velocity-independent scheme of Hellmer and Olbers (1989) that assumed a constant ocean velocity 15 cm/s. It should be noted that using the velocity-independent parameterization also implies that we ignore the local enhancement in melt rates due to the formation of buoyant upslope plumes arising from the upstream ice-shelf melting. However, such effects are considered secondary compared to the primary influence of the dominant heat supply through warm water inflow into the ice-shelf cavities.

**L620–629 (in Summary and Discussion section)**

For instance, the accuracy of the bathymetric data underneath ice shelves can greatly affect the representation of circulation under the ice shelf. The ocean velocity at the ice-shelf bases has an impact on both the patterns and magnitude of ice-shelf basal melting. This effect is especially pronounced when a velocity-dependent parameterization is used for thermal and salinity exchanges at the ice-ocean interface (Mueller et al., 2012; Dansereau et al., 2014). It's worth noting, however, that this study employed a velocity-independent parameterization. While this approach captures the ice-shelf basal melting driven primarily by the dominant thermal forcing originating from the mCDW intrusion, it cannot account for secondary effects like enhanced melting patterns due to buoyancy-driven upslope flows generated by the upstream ice-shelf meltwater. In scenarios where velocity-dependent parameterization is used, the melting pattern in the model might change in regions with stronger currents, offering a potentially more intricate view of the ice-ocean interactions. Incorporating the updated datasets,…

> **(2)** *L485-500: The analysis of inflow and outflow transport across the southern boundary of the Slope Box is missing (Fig. 18). The Slope Box is different from the Sabrina Depression box since the Slope Box has an open southern boundary. Therefore, the transport balance between the inflow and outflow of the Slope Box can not be explained by the calculation confined within the western, northern, and eastern boundaries. The author may add the calculation and description of inflow and outflow at the southern boundary of the Slope Box in Fig. 18.*
>
> **(3)** *L490: The authors calculate the inflow and outflow from the surface to 800 m.*
>
> *Does the vertical transport across the bottom boundary at 800 m depth have some influence on the balance of the inflow and outflow? It would be nice if the authors could have a short discussion on this.*
>
> *Technical Corrections:*
>
> **(4)** *L115: 'there remains large uncertainties' should be 'there remain large uncertainties'*

**(2)** Thank you for pointing out the need to analyze the inflow and outflow transport across the southern boundary of the Slope box. In this revision, we have added a description noting that the outflow and inflow across the southern boundary of the Slope box essentially mirror the flows across the northern boundary of the SD box. Furthermore, for clarity, we have included the boundaries of both the Slope and SD boxes in the insets of the corresponding figures.

**L465–468**

The western, eastern, and northern boundaries of the Slope box were set to 117° E, 121° E, and the 2500-m depth contour, respectively. The Slope box's southern boundary shares a large part of the SD box's northern boundary, and thus the inflow and outflow across the Slope box's southern boundary corresponds to the outflow and inflow across the SD box's northern boundary (Fig. 9e and 9b), respectively.

**(3)** As you pointed out, the vertical transport across the 800-m interface contributes to the total water mass balance of the Slope box. A simple estimation based on the annual-mean transport balance indicates that there should be downward transport of approximately 250 mSv across the 800 m interface. However, the results of the downward transport would be sensitive to the control box definition. In the revised manuscript, we have added the reason why we focus on the lateral inflows and outflows.

**L468–475**

We only calculated the inflow and outflow transports from the surface to 800 m, to focus on water mass exchange across the shelf break, which has a depth of less than 650 m. On an annual-mean basis, there is a substantial inflow from the eastern boundary into the Slope box, with the total transport over 1000 mSv (Fig. 12c). Balancing this annual-mean inflow transport, there are outflow transports exceeding 400 mSv at both the western and northern boundaries (Fig. 12d–e), southward transport to the SD box (Fig. 9b and 9b), and downward transport across the 800 m interface. As shown later (Figs. 13 and 14), offshore water flowing across the shelf break to the continental shelf region resides on the upper continental shelf. Therefore, in this section, we focus on the lateral inflow and outflow patterns. The lateral inflow/outflow pattern …

**(4)** We have corrected it (L117).

> **(5)** *L120: 'It has been suggested that the inflow of mCDW onto continental shelf regions is related to the Antarctic Slope Front/Current (ASF/ASC) system on the upper continental slope region (Nakayama et al., 2021; Thompson et al., 2018; Silvano et al., 2019)'.*
>
> *The study of Liu et al. (2013) described the dynamic mechanisms responsible for mCDW intrusions regulated by the ASC/ASF system in East Antarctica, and it might be suitable to be cited here.*
>
> *Liu, C., Z. Wang, X. Liang, X. Li, X. Li, C. Cheng, and D. Qi, 2022: Topography-Mediated Transport of Warm Deep Water across the Continental Shelf Slope, East Antarctica. J. Phys. Oceanogr., 52, 1295–1314, https://doi.org/10.1175/JPO-D-22-0023.1.*
>
> **(6)** *L395: 'with the seasonal peaks occurring from May to September in the TIS and eTIS' and 'The wMUIS reaches its peak between April and May, while the MUIS reaches its peak between October and December'*
>
> *It would be nice if the authors could also identify these periods directly in Fig. 13 by using boxes or something else.*
>
> **(7)** *L445: 'with the remaining 0.1% being explained by the sum of ice-shelf basal melting from the southern boundary (i.e., ice-shelf fronts), sea-ice production, transport, and melting over the SD box'*
>
> *Is the ocean model used in this study a volume-conserved model? If the ocean model is volume-conserved, the ice shelf basal melting and the sea ice evolution only change the salinity rather than the volume transport. The remaining '0.1%' may be attributed to the truncation error in the volume transport calculation, and it is so small that such remaining can be omitted without particular attention.*
>
> **(8)** L555: 'The southward heat transport timeseries' should be 'The southward heat transport time series'
>
> **(9)** *L575 and L675: 'This means that a positive value in the SAM index leads to weaker coastal winds.'*
>
> *It would be nice if the authors could add some references corresponding to the weaker coastal winds in a positive SAM index.*
>
> **(10)** *L635: 'in the surface layer, but this study' should be 'in the surface layer, this study'*
>
> **(11)** *L640: 'The present model results shows that' should be 'The present model results show that'; 'at mid depths' should be 'at mid-depths'*
>
> **(12)** *L650: 'From modeling perspective,' should be 'From a modeling perspective,'*
>
> **(13)** *L655: 'where the Antarctic Circumpolar Current (ACC) deflect southward' should be 'where the Antarctic Circumpolar Current (ACC) deflects southward'*
>
> **(14)** *L1095: The labels of the vertical overturning circulation in Fig 14b are missing.*

**(5)** We have added the reference (L125).

**(6)** In the corresponding figure (Fig.7 in the revised manuscript), in order to highlight the seasonality, we have added dashed line showing the annual-mean volume/heat transport of water masses denser than 27.60 kg m$^{-3}$.

**(7)** The sentences were removed not to stray from the main storyline as well as to shorten the manuscript. As you pointed out, the model is volume-conserved one, but the ocean component exchanges the freshwater with the sea-ice and ice-shelf components.

**(8)** and **(10–13)** We have corrected them.

**(9)** We have added a reference about weakening of Antarctic coastal wind.

Neme, J., England, M. H., and McC. Hogg, A.: Projected Changes of Surface Winds Over the Antarctic Continental Margin, Geophys. Res. Lett., 49, https://doi.org/10.1029/2022GL098820, 2022.

**(14)** We have added labels of the contours in Fig 8 (Fig. 14 in the old manuscript).

**Response to the comments from Reviewer #2**

(NB *italicized text in box* is comments from the reviewer.)
* * *
**GENERAL COMMENTS**

*The manuscript fits within the scope of the journal. By bringing a long-term perspective (70 years) based on a new bathymetric compilation that is considerably more realistic than in earlier modeling studies, it represents substantial progress beyond our current scientific understanding of this region. The purpose of the work is clearly articulated in the text, I did not find substantial flaws in the methodology, and the conclusions of the study are adequately supported by the figures/tables of the manuscript. The results of the manuscript are appropriately discussed in the context of the existing literature and the authors abundantly cite other related work. Overall, this is a substantial contribution providing considerable detail on the melt of ice shelves in a region storing the equivalent of 3.5m of global sea level, which is obviously relevant to the large fraction of the population living along the world's coastlines.*

**SPECIFIC COMMENTS : MAJOR**

*(1) The one major flaw that I'm noticing is that the authors apparently made no effort in limiting the number of figures and tables (24 figures, 3 tables, which is something I've never seen before). So the manuscript fails the journal's requirement of "scientific results and conclusions presented in a clear, \*concise\*, and well-structured way". This being said, if the authors are reasonable, it should be easy to address this problem by following some, or all, of the following suggestions:*

*(a) This journal allows for a "Supplement", and I would encourage the authors to use it extensively. While the new bathymetric compilation is a major outcome of the study, I would think that Figure 2 represents more information than the average reader is interested in seeing, and so it could be moved to "Supplement". Figure 4 is mostly methodological in nature and could be moved to "Supplement". Figures 5-6 demonstrate the realism of the model but they are not necessary to support the scientific results (and therefore Figs.5-6 could go to "Supplement"). The timeseries of Fig.9 look so much like a sinusoid that one could say that they are already appropriately described by the panels inside Fig.8 (making Fig.9 a bit redundant). Figure 11 is barely discussed in the text (most of the text about Fig.11 is spent describing the Methodology behind it) and could be moved to "Supplement" as far as I'm concerned. Figures 19-22 appear (to me) to describe the same story in different ways, with Figs.19-20 being only very briefly mentioned in the text; given that, it's hard to believe that all of Figs.19-22 deserve to appear in the body of the manuscript.*

*(b) Couldn't Tables 2-3 be combined together? This would facilitate a comparison between this study (Table 2) and the earlier studies (Table 3).*

*(c) Another possibility is to use "(not shown)" when a result is interesting enough to be mentioned but it isn't critical to the scientific demonstration. For example, I felt like the 8 year cycle apparent in Fig.11(a) was a bit interesting and worth briefly mentioning in the text, but I didn't feel the need to see it as one of the manuscript's figures.*
* * *
**(General comments)**

Thank you very much for your careful reading of our manuscript and your constructive comments. We are pleased to hear that you find our work interesting, and that you recommend publication in TC. Your feedback has been invaluable for improving the quality of our paper, and we are committed to addressing your comments and suggestions.

**(1-a) (1-c)** Following your suggestion, we have shortened the manuscript by moving several figures to Supplement. Figures 4, 5, 6, 9, 11, and 22 in the previous manuscript were moved to Supplement, and Figure 2 was moved to the Appendix. In the following response, we use new figure numbers in the revised manuscript.

**(1-b)** Following your suggestion, we have merged the tables.

**SPECIFIC COMMENTS : MINOR**

**(2)** *The journal requires the "experiments and calculations" to be "sufficiently complete and precise to allow their reproduction by fellow scientists (traceability of results)". Based on the authors' "Data availability" statements, they plan to share the model results supporting their conclusions "after acceptance". So I wasn't able to access the model results, but the authors seem to be willing to share them, eventually. The editor can decide whether this is good enough (or not).*

**(3)** *Another requirement of the journal is that the language be "fluent and precise". Overall, I thought the whole manuscript was well-written. My main concern is that some portions of the text felt unecessarily long. For example, lines 45-75 are fairly general and would fit better in an introductory paper (or a graduate student's thesis) than in a research paper where conciseness is key. Another example is line 591-616, which acts as a repetition of what the reader already saw inside Sections 3-6. I personally don't see the need for lines 591-616; the Abstract already provide a summary of the key results.*

**(4)** *Line 52: "...and iceberg caving at ice-shelf fronts..." (Typo: "caving")*

**(5)** *Line 89: "Previous observational and modeling studies have inferred that the interannual variability in glacier/ice sheet variable is strongly controlled..."*
*Something is wrong in the sentence; maybe delete the word "variable"?*

**(6)** *Line 127: "...heavily influenced by their artificial lateral boundaries, the conditions which were often derived from a different coarse-resolution ocean model."*

*The sentence could be interpreted by a reader as "all regional models are flawed by having artificial boundaries", which would be inaccurate. (The literature is filled with successful regional model implementations using highly-realistic lateral conditions.) Removing the word "artificial" from the sentence would make it a lot better; it gives more weight to the second part of the sentence and ties the criticism to something specific and quantifiable (the coarseness of the oceanic dataset used for the lateral conditions).*

**(7)** *Section 3 (Sea-ice extent and production): I could not find where the "observed" "sea ice concentration and extent" were from. What datasets were used? SSMI? Something else? This needs to be specified.*

**(2)** We have uploaded data to a data repository (Mendeley Data). Please find the link in the revised manuscript.

**(3)** Thank you for your thoughtful suggestions regarding the length and structure of the "Introduction" and "Summary and Discussion" sections. We deeply considered your advice. However, we have decided to retain the sections as they are for the following reasons. Given that The Cryosphere encompasses a broad spectrum of topics within the cryosphere, our intention in the Introduction is to navigate a diverse readership from general concepts to the specific topics of our study. As for the "Summary and Discussion" section, it serves to provide more detailed correspondence with the figures presented in the paper, which we feel is not fully covered by the Abstract alone.

**(4), (5),** and **(6)** We have corrected them.

**(7)** We have added the following sentence in the sea-ice section.

**L251–253**

Figure S2 shows the observed and modeled seasonal cycle of regional sea-ice concentration and extent from 108°E to 128°E. We used observed sea-ice concentration derived from satellite passive microwave data using the NASA team algorithm (Cavalieri et al., 1984; Swift and Cavalieri, 1985).

**(8)** *Line 393: "The upper panels in Fig.13 show the net inflow volume transports..."*

*The word "net" is doing more harm than good in this sentence. "Net" is often used to represent "inflow minus outflow", while in this case the authors are really referring to the inflow. I assume you added "net" to refer to the fact that the inflow was summed across the ice shelf front, but I believe it is creating more confusion than anything. Could you take out the word "net"?*

**(9)** *Line 514-516: "We calculated the transports of the surface current by integrating the westward transport in density layers lighter than 27.7 kg/m3 and that of the undercurrent by integrating the eastward transport in density layers denser than 27.6 kg/m-3."*

*Why would you have such an overlap in your definition of the surface/undercurrent? As things stand, the density range 27.6-27.7 kg/m3 is accounted in both the surface and the undercurrent. Is it because these two currents reach their maximum magnitude at different times in the seasonal cycle, and the overlap was necessary to capture the full magnitude of the surface/undercurrents at those moments? Please clarify.*

**(10)** *Caption of Figures 7, 15: In an ideal world, the reader would be able to look at the figures and understand them without having to read the full manuscript. In the caption, could you replace the acronym "SD box" by "Sabrina Depression (SD) box"?*

**(11)** *Caption of Fig.10: Please define acronym "CKDRF" in the caption; this particular acronym only shows up a few times in the whole manuscript, so we shouldn't expect the reader to know or remember what it stands for.*

**(8)** This is correct with "net". As you correctly understood, we calculated the difference between the inflow and outflow transport in each bin, and accumulated it when the net inflow is positive.

**L370–373**

The volume transports and the mean temperatures were calculated for each 0.02 kg m$^{-3}$ bin of the potential density for the inflow and outflow components. Net inflow transport was determined only when the difference between inflow and outflow transport was positive. The upper panels in Fig. 7 show the net inflow volume transports in each density bin and the mean temperatures of the inflow component.

**(9)** Based on the vertical structure of the east-westward velocity observed in Figure 15, we identified the density layer of 27.6–27.7 kg m$^{-3}$ as a transition zone. This is because the upper portion of the undercurrent can also be observed within this transition zone. Since the direction of the velocity has been accounted for in our calculations, this overlap does not introduce any errors in our calculations for volume and heat transports. We have added the sentence to clarify the transition layer.

**L495–498**

We calculated the transports of the surface current by integrating the westward transport in density layers lighter than 27.7 kg m$^{-3}$ and that of the undercurrent by integrating the eastward transport in density layers denser than 27.6 kg m$^{-3}$. We treat the density layer in 27.6–27.7 kg m$^{-3}$ as a transition layer between the ASF and the seasonally-formed undercurrent (Fig.15). The surface current transport becomes …

**(10) and (11)** Following your suggestion, we have modified them.

(12) *Figure 10: As the authors point out in their manuscript, this is one of the first studies to provide a multi-decadal (70 years) perspective on glacial ice loss, so there's an opportunity to provide new insight to the community. If we assume the 69-72.6 Gt/yr grounding line flux of Rignot et al. 2019 is valid over the period 1979-2017, and that the modeled basal melt of Fig.10 is accurate, then what fraction of the grounding line flux is due to calving? How does this fraction compare to the one reported in Fig.1 of Rignot et al. 2013 (although the time periods are different)?*

*Similarly, if we assume that the calving is constant over time, how does the trend apparent in Fig.10a over 2008-2018 compare with the trends in Miles et al. 2022 over the same period?*

(13) *Caption of Figure 12: Something is wrong in the sentence: "...velocity along a 5-km off along a line section of...".*

(14) *Caption of Figures 12, 17: Specify the units for potential density anomaly (kg/m3).*

(15) *Caption of Figure 13: See my earlier comment about "net".*

(16) *Caption of Figure 14: Clarify that the stream function is computed from laterally-integrated velocities, clarify that the units for the stream fucntion are mSv.*

(17) *Caption of Figure 24: Mention what geographical area the "wind speed" is representative of (presumably the yellow box of Figure 3, but this needs to be clarified).*

**(12)** We consider that estimating the calving volume/fraction is beyond the scope of this study, as it requires consideration of the dynamical processes of ice sheets and ice shelves. Following you suggestion, we have added comments about comparison with recent oceanographical and ice-sheet studies.

**L571–581**

Importantly, the model has a long-term integration spanning more than 70 years making the output suitable for examining seasonal, interannual to decadal variability. While a direct comparison between our model's TIS basal melting (Fig. 5a) and observational data is challenging, we find our model's interannual variability to be consistent with observed variability and trend, including the temperature rise over the continental shelf from 2015-2022 (Rintoul et al., 2016; Hirano et al., 2023), the decrease in ice discharge after 2010 within the 2008–2018 observation period (Miles et al., 2022), and the long-term TG acceleration since the 1960s (Li et al., 2023). Constraints and uncertainties in our model, such as the assumption of a steady-shape ice shelf, sub ice-shelf topography, and ice-ocean parameterization, as well as uncertainties like the time lag in ice sheet/shelf response to ocean forcing through basal melting, complicate a direct comparison across the different variables. However, within these limitations, our findings do not contradict existing observations and provide new insights into interannual-to-decadal variability.

**(13)** We have corrected it.

**caption in Fig. 6**

Vertical profiles of (a) potential temperature, (b) salinity, and (c) velocity along a line section located 5 km off the Sabrina Coast.

**(14) and (17)** We have modified the captions.

**(15)** Please refer to the reply to (8).

**(16)** We have added the explanation in the caption.

The stream function is calculated by integrating the volume transport laterally along the same distances from the ice-shelf within the same density layers and then accumulating from the dense to light water masses.